# General Scene Adaptation for Vision-and-Language Navigation

**Haodong Hong**[12†], **Yanyuan Qiao**[3], **Sen Wang**[1], **Jiajun Liu**[21], **Qi Wu**[3*]
[1]The University of Queensland, [2]CSIRO Data61, [3]The University of Adelaide
{haodong.hong, sen.wang}@uq.edu.au, jiajun.liu@csiro.au
{yanyuan.qiao, qi.wu01}@adelaide.edu.au

## Abstract

Vision-and-Language Navigation (VLN) tasks mainly evaluate agents based on one-time execution of individual instructions across multiple environments, aiming to develop agents capable of functioning in any environment in a zero-shot manner. However, real-world navigation robots often operate in persistent environments with relatively consistent physical layouts, visual observations, and language styles from instructors. Such a gap in the task setting presents an opportunity to improve VLN agents by incorporating continuous adaptation to specific environments. To better reflect these real-world conditions, we introduce GSA-VLN (General Scene Adaptation for VLN), a novel task requiring agents to execute navigation instructions within a specific scene and simultaneously adapt to it for improved performance over time. To evaluate the proposed task, one has to address two challenges in existing VLN datasets: the lack of out-of-distribution (OOD) data, and the limited number and style diversity of instructions for each scene. Therefore, we propose a new dataset, GSA-R2R, which significantly expands the diversity and quantity of environments and instructions for the Room-to-Room (R2R) dataset to evaluate agent adaptability in both ID and OOD contexts. Furthermore, we design a three-stage instruction orchestration pipeline that leverages large language models (LLMs) to refine speaker-generated instructions and apply role-playing techniques to rephrase instructions into different speaking styles. This is motivated by the observation that each individual user often has consistent signatures or preferences in their instructions, taking the use case of home robotic assistants as an example. We conducted extensive experiments on GSA-R2R to thoroughly evaluate our dataset and benchmark various methods, revealing key factors enabling agents to adapt to specific environments. Based on our findings, we propose a novel method, Graph-Retained DUET (GR-DUET), which incorporates memory-based navigation graphs with an environment-specific training strategy, achieving state-of-the-art results on all GSA-R2R splits. The dataset and code are available at `https://github.com/honghd16/GSA-VLN`.

## 1 Introduction

Vision-and-Language Navigation (VLN) (Anderson et al., 2018b) aims to enable agents to navigate to a specific destination following language instructions. Traditional VLN researches (Qi et al., 2020; Thomason et al., 2020) mainly focus on evaluating agents using strictly unseen instructions and environments to assess their generalization capabilities. This "unseen" criterion applies not only to the data between training and evaluation phases, but also to individual evaluation instances, where agents are tested on each instruction-trajectory pair without prior knowledge of the environment.

However, this setting diverges from practical navigation situations. In real-world applications, such as indoor household robots, agents actually operate in a consistent environment over time. As they execute more instructions, the working environment generally transitions from "unseen" to "seen" and agents become increasingly familiar with the environment. This leaves room for agents to adapt and improve their performance through environmental adaptation, which can significantly

---

*Corresponding author. †Work done during internship at Australian Institute for Machine Learning (AIML).

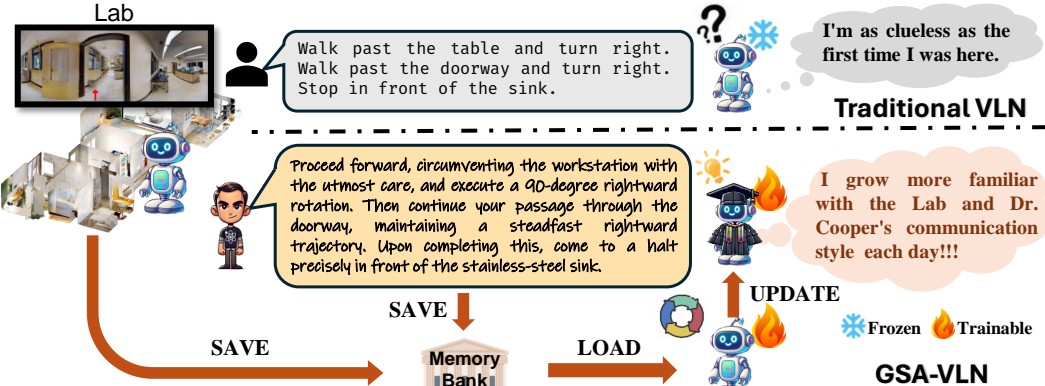

Figure 1: Comparison between the traditional VLN task and the proposed GSA-VLN task. Traditional VLN agents can only execute fixed-style instructions with frozen parameters, remaining unfamiliar with the environment like the laboratory even after extended use. In contrast, GSA-VLN enables agents to dynamically update parameters, leverage long-term history from the memory bank, and quickly adapt to both the environment and varying instruction styles from different users.

enhance the agent performance of existing VLN methods. IVLN (Krantz et al., 2023) emphasizes the persistent environment problem by linking paths of each scan into long-horizon tours, but the limited number of paths (6-100) restricts its applicability to optimization-based methods like Test-time Adaptation (TTA) (Liang et al., 2024) and fails to replicate real-world usage scenarios.

Therefore, we propose a novel task called General Scene Adaptation for VLN (GSA-VLN). As shown in Fig. 1, in this task, agents are required to maintain long-term memory and continuously update their model parameters to improve performance over time while executing navigation instructions in a specific scene throughout their lifetime. Externally, the agent behaves the same as traditional VLN settings. However, internally, it evolves during this process by adapting to the working environment without additional feedback or assistance, a process known as unsupervised learning, making our task both practical and realistic.

Although GSA-VLN focuses on environment-specific adaptation, the agent must also be general enough to adapt to a diverse range of scenes as its target environment, given the wide variety of real-world settings. However, the limited number and diversity of environments and instructions in existing VLN datasets raise concerns about whether the trained agents can be deployed across various real-world situations. Currently, all indoor VLN tasks are built on the Matterport3D dataset (Chang et al., 2017) with fewer than 30 evaluation buildings, most of which are residential. While some works expand the scope by incorporating additional datasets (Chen et al., 2022b; Wang et al., 2023b) or generating synthetic ones (Li et al., 2022; Li & Bansal, 2024), they are only used for training rather than evaluation. Moreover, instructions in these tasks are typically concise and simplistic, lacking personal characteristics or distinctive language habits that reflect real-world user interactions. Since both the environment and instructions are in-distribution (ID) with the training data, it is essential to evaluate agents on out-of-distribution (OOD) data to ensure broader generalization.

To this end, we introduce the GSA-R2R dataset, which provides a comprehensive collection of environments and instructions for evaluating agent performance in both ID and OOD contexts within a single scene. We incorporate buildings from the Habitat-Matterport3D (HM3D) dataset (Ramakrishnan et al., 2021), which offers a broader range and greater number of photorealistic environments compared to the MP3D dataset, supporting more general evaluation. Since most training environments are residential, we categorize residential buildings as ID data and conduct building-scale classification to identify non-residential structures, such as cinemas and shops, as OOD data, resulting in a diverse set of 150 evaluation buildings with various layouts and structures.

Besides visual environments, instruction variety is a crucial factor in scene adaptation, as instructions for a specific environment are often consistent in style, reflecting the unique speaking patterns of the instructors. For example, household agents are typically instructed by homeowners, while school-based agents are primarily used by teachers and students. These instructions are delivered in a consistent style, reflecting either personal speaking habits or the professional language associated with the roles in the building. However, this aspect has been largely overlooked in current VLN research. To address this, we develop a three-stage instruction orchestration pipeline to pro-

duce abundant and diverse instruction-trajectory pairs for each environment. Initially, we use a trained speaker model (Tan et al., 2019) to produce noisy instructions, which are then refined by large Vision-Language Models (VLMs)(Achiam et al., 2023) with path visualizations, and finally rephrased by Large Language Models (LLMs)(Achiam et al., 2023) to simulate diverse instructional styles of specific characters or typical users of the building type. As a result, each environment is equipped with 600 instructions across multiple speaking styles, providing a more realistic and valuable resource for future VLN research.

We conduct a comprehensive evaluation of our generated data using both automatic metrics and human studies to validate its quality and diversity. We also test various existing adaptation methods, revealing that unsupervised training approaches are ineffective in our setting, while explicitly maintaining the memory of previous visual and textual information shows promise in improving agent performance in GSA-R2R. Based on these findings, we propose a novel method called Graph-Retained DUET (GR-DUET), which continuously updates an overall topological graph for each environment to preserve historical information in both training and evaluation. This method achieves an 8% improvement in Success Rate (SR) compared to the vanilla DUET (Chen et al., 2022c) and produces state-of-the-art results across all splits.

## 2 RELATED WORK

**Vision-and-Language Navigation (VLN).** VLN tasks involve agents following natural language instructions to reach a specified target (Gu et al., 2022; Zhang et al., 2024), a challenge first introduced with the Room-to-Room (R2R) dataset (Anderson et al., 2018b). Since then, numerous indoor VLN datasets emerged (Krantz et al., 2020; Jain et al., 2019), which mainly focus on varying the textual inputs, such as high-level object-oriented instructions (Qi et al., 2020; Zhu et al., 2021), multilingual instructions (Ku et al., 2020), and multi-modal instructions (Hong et al., 2024). However, they are all based on the same scenarios from the Matterport3D dataset (Chang et al., 2017). Many works propose to introduce novel scenarios by incorporating other datasets (Chen et al., 2022b; Wang et al., 2023b) or utilizing web data (Lin et al., 2023). However, all these methods are only used as additional training data with evaluations still being conducted on the original splits with limited diversity. In contrast, we include diverse environments and instructions in the evaluation splits to fully evaluate agent adaptability in both ID and OOD contexts.

**Adaptation Methods in VLN.** Although no prior works in VLN have addressed the problem of single-scene adaptation, some studies offer potential solutions, such as the pre-explore setting (Wang et al., 2019). We categorize them into two kinds. The first is optimization-based methods, where the parameters of the navigation model are updated within the target environment. TTA methods like TENT (Wang et al., 2021) and SAR (Niu et al., 2023) further optimize the model parameters with the objective of entropy minimization, while Back-Translation (Wang et al., 2020) uses a trained speaker to generate instructions for imitation learning. The second is memory-based methods, which explicitly store information about seen places and instructions to help decision-making, as seen in methods of IVLN (Zhao et al., 2024) and RREx-BoT (Sigurdsson et al., 2023). For example, TourHAMT (Krantz et al., 2023) incorporates previous episodes as additional history embeddings, while OVER-NAV (Zhao et al., 2024) detects keywords from instructions within observations and stores the results in an Omnigraph to assist navigation. Similarly, SG-Nav (Yin et al., 2024) constructs consistent 3D scene graphs to represent environments, offering a powerful approach for recording and utilizing historical information. Recently, LLM-based VLN methods, such as InstructNav (Long et al., 2024) and NavCoT (Lin et al., 2024), have demonstrated strong zero-shot navigation performance, highlighting their potential to address the scene adaptation problem.

**Persistent Environments in VLN.** The concept of agents operating in persistent environments has led to several popular tasks in embodied AI, such as multi-object navigation (Wani et al., 2020) and multi-target embodied QA (Yu et al., 2019). Iterative VLN (IVLN) (Krantz et al., 2023) introduces this idea to VLN by organizing all instructions in sequence to form a long-horizon tour and enable memory throughout the tour. Our task differs from IVLN in two key aspects. First, while both tasks focus on enhancing agent performance within a single environment, IVLN emphasizes long-horizon navigation with a single tour per environment, whereas our GSA-VLN focuses on enabling agents to adapt to each environment from a diverse range of visual buildings and instruction types. Second, IVLN only includes a limited number of trajectory-instruction pairs for each environment, making it not suitable for optimization-based ones.

# 3 TASKS AND DATASETS

## 3.1 PRELIMINARIES

In VLN task, the agent is required to follow a given natural language instruction $X = (x_1, x_2, ..., x_L)$ consisting of $L$ words to navigate to the target destination. Specifically, the agent is initially placed at the start node $v_0$. At each time step $t$, the agent predicts an action $a_t$ to move to one of the neighboring nodes in the connectivity graph $\mathcal{G}$ of the environment, guided by the visual observation $O_t$, the language instruction $X$, and the history of previous steps $H_t = \{O_0, a_0, O_1, a_1, \cdots, O_{t-1}, a_{t-1}\}$. Typically, the observation $O_t$ is represented as a panorama composed of $N = 36$ discrete views $O_t = \left\{o_t^i\right\}_{i=1}^{N}$. This process continues until the agent either reaches the predefined step limit or selects the special [STOP] action.

## 3.2 THE GSA-VLN TASK

Most VLN tasks evaluate each instruction independently, initializing the history as empty ($H_0 = \emptyset$) and keeping the model parameters fixed ($\theta = \theta_0$). While this setting effectively assesses the agent's ability to interpret and follow isolated instructions, it fails to capture the continuity and adaptation required in real-world navigation scenarios. In practical applications, both environments and instruction styles remain consistent over time, enabling agents to accumulate and leverage contextual knowledge to enhance performance. To address this limitation, we propose the GSA-VLN task to introduce the challenge of single-scene adaptation, enabling agents to continuously improve as they execute instructions in previously unseen environments.

Specifically, GSA-VLN introduces an environment-specific memory bank $\mathcal{M}_E$, which stores historical information from all executed episodes within a given environment $E$. This memory bank dynamically expands as the agent executes instructions, capturing four key components: visual observations ($\mathbf{O}$), the instructions ($X$), selected actions ($\mathbf{A}$), and trajectory paths ($\mathbf{P}$). For example, after executing $k$ instructions in environment $E$, the memory bank is updated as follows:

$$\mathcal{M}_E = \{X^{1:k}, \mathbf{O}^{1:k}, \mathbf{A}^{1:k}, \mathbf{P}^{1:k}\} \tag{1}$$

Externally, agents in GSA-VLN behave similarly to those in standard VLN tasks when executing instructions. However, internally, the agent can leverage the memory bank to adapt to the current working environment for better performance, depending on the method employed. The stored memories in the memory bank represent the execution history of agents, although there may exist misalignment between instructions and paths due to navigation errors, all the memories are treated as unlabeled data and are primarily used for unsupervised learning techniques.

There are two primary distinctions between GSA-VLN and standard VLN tasks. First, the agent can access the memory bank to retrieve long-term history, $H_0 = \mathcal{M}_E' \subseteq \mathcal{M}_E$, instead of beginning each navigation episode without prior knowledge:

$$a_0 = \pi(O_0, X, H_0; \theta) \tag{2}$$

Second, the parameters of agents can be updated during the navigation process by employing unsupervised learning techniques on data from the memory bank:

$$\theta' = \theta - \alpha \nabla_\theta \mathcal{L}(\mathcal{M}_E, \theta), \tag{3}$$

where $\theta$ denotes the parameters of the navigation model. Notably, While GSA-VLN aims to develop environment-specific agents $\theta'$, the initial model $\theta_0$ should be general enough to be applied to various environments, maintaining a level of environment-agnosticism:

$$\max_{\theta_0} \mathbb{E}_{E \sim \mathcal{E}} \left[ \mathcal{P}(E; \theta'(\theta_0)) \right] \tag{4}$$

where $\mathcal{E}$ represents the environment distribution, and $\mathcal{P}(E; \theta'(\theta_0))$ denotes the agent performance in environment $E$ with updated parameters $\theta'$, which are adapted from the initial parameters $\theta_0$.

We address differences between GSA-VLN and the related areas. Unlike lifelong learning (Liu, 2017), which focuses on acquiring multiple skills over time, our work emphasizes repeated mastery of the same navigation skill within a scene-specific context. While TTA (Gao et al., 2024) adapt an agent during inference without supervision, our approach extends TTA by integrating a dynamically updating memory bank, enabling fixed-parameter adaptation with varying inputs.

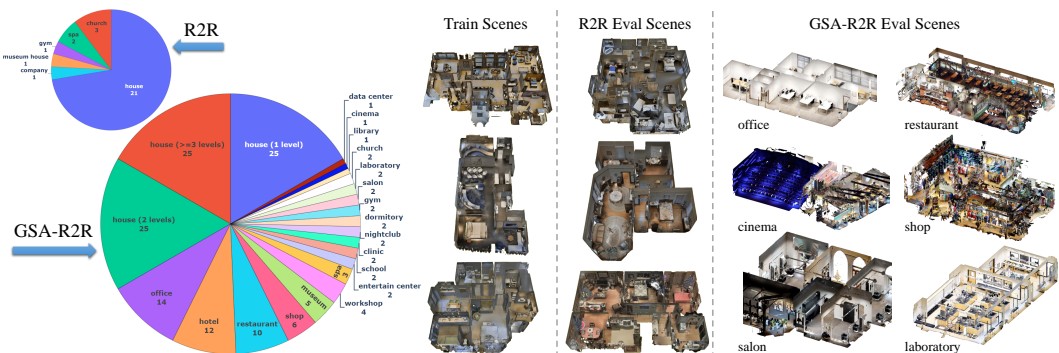

Figure 2: **Left**: Building type counts in R2R and GSA-R2R. **Right**: Comparison of buildings in R2R and GSA-R2R. Unlike R2R, where evaluation scenes are similar to the training set, GSA-R2R includes a more diverse mix of both in-distribution (ID) and out-of-distribution (OOD) data.

## 3.3 THE GSA-R2R DATASET

The limited number and diversity of existing VLN datasets make them unsuitable for the GSA-VLN task. For instance, the most widely used VLN dataset, R2R (Anderson et al., 2018b), includes 90 building-scale scenes from the Matterport3D (MP3D) dataset (Chang et al., 2017), with only 29 scenes for evaluation and most of them are residential houses. Each building contains a limited number of paths, ranging from a maximum of 100 to as few as 6, with each path paired with three natural language instructions that share a similar concise and plain style. Therefore, we propose the GSA-R2R dataset, which not only provides sufficient data to allow continuous agent optimization to test their **specialization**, but also includes a diverse range of building types and instruction types to evaluate the agent **generalization** to various application scenarios.

### 3.3.1 ENVIRONMENT

The Habitat-Matterport3D (HM3D) dataset (Ramakrishnan et al., 2021) has 800 photorealistic buildings with more diverse building types and structures compared to MP3D. Therefore, we choose to incorporate HM3D environments into agent evaluation to assess whether agents can adapt to more diverse settings. We categorize the buildings into two types: residential and non-residential, with the former as ID data and the latter as OOD data since the training data are most residential ones.

Specifically, we collect and manually refine 187 building types from the OpenStreetMap website [1], which provides crowd-sourced building tags worldwide. For each environment, we use GPT-4 (Achiam et al., 2023) to predict the building type from these categories based on three types of image prompts: an overview of the building, top-down views of each floor, and panoramas selected using spectral clustering (Ng et al., 2001) on the graph nodes to capture representative visual observations. For non-residential results, we manually verify and correct the predictions and apply two filtering rules. First, we exclude environments in the R2R training set to avoid data leakage, ensuring that existing VLN models could be directly evaluated using our dataset. Second, we remove scans with fewer than 600 potential paths to ensure a sufficient number of paths per scan. In total, we identify 75 non-residential buildings

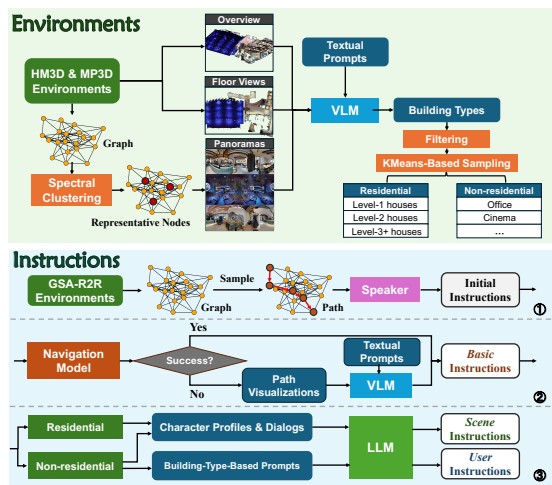

Figure 3: The generation procedure of the GSA-R2R dataset. **Up**: environment selection. **Bottom**: the three-stage instruction orchestration pipeline.

across 19 different types with 70 from HM3D and 5 from MP3D. To balance the dataset, we sample

---

[1]https://taginfo.openstreetmap.org/keys/building

Table 1: Compared to the evaluation part of existing datasets in embodied navigation tasks. †: TouchDown is an outdoor dataset in New York City, which makes it hard to define scene types.

| Dataset | Source | Scenes | | Path | Instructions | | Vocab Size | |
|---|---|---|---|---|---|---|---|---|
| | | Num | Type | Num | Num | Type | All | Unseen |
| R2R (Anderson et al., 2018b) | MP3D | 29 | 6 | 2,174 | 6,522 | 1 | 1,946 | 545 |
| R4R (Jain et al., 2019) | MP3D | 29 | 6 | 5,026 | 45,234 | 1 | 1,230 | 221 |
| RxR-en (Ku et al., 2020) | MP3D | 29 | 6 | 4,201 | 8,636 | 1 | 4,789 | 1,387 |
| CVDN (Thomason et al., 2020) | MP3D | 29 | 6 | 2,741 | 2,741 | 1 | 1,559 | 490 |
| TouchDown† (Chen et al., 2019) | Google Street View | 1 | - | 2,800 | 2,800 | 1 | 3,104 | 759 |
| RobustNav (Chattopadhyay et al., 2021) | ROBOTHOR | 15 | 1 | 1,800 | - | - | - | - |
| PASTURE (Gadre et al., 2023) | ROBOTHOR | 15 | 1 | 2,520 | 2,520 | 1 | 123 | 111 |
| GSA-R2R (Ours) | MP3D & HM3D | 150 | 20 | 90,000 | 90,000 | 7 | 4,337 | 2,905 |

an equivalent number of residential buildings and classify them into three categories based on the number of levels. For each category, we use K-Means clustering to select 25 scans based on metadata such as room count and navigable areas to cover a diverse set of layouts and structures within the 75 residential buildings, with 65 from HM3D and 10 from MP3D, as shown in Fig. 2.

### 3.3.2 INSTRUCTION

With the selected environments, we employ the navigation graphs from ScaleVLN (Wang et al., 2023b) and randomly sample 600 paths for each environment. To simulate the diverse speaking styles in real-world scenarios, we develop a three-stage instruction orchestration pipeline to produce three types of instructions for these sampled paths, as illustrated in Fig. 3.

In the first stage, we employ the EnvDrop speaker (Tan et al., 2019) trained on R2R to generate initial instructions[2]. However, some of these instructions contain noise and inaccuracies, requiring further refinement. Therefore, in the second stage, we utilize a navigation model trained on the unselected paths from the 150 environments to identify incorrect instructions by using successful execution as an indicator of instruction feasibility. For failed instructions, we leverage GPT-4 to detect misalignments between trajectories and instructions,and correct them using specialized path visualization prompts, resulting in refined *"Basic"* instructions. Building on the *"Basic"* instructions, we further rephrase them in the third stage to reflect distinct speaking styles. For non-residential buildings, we use an LLM to select a potential user identity based on the scene type and rephrase the instructions into *"Scene"* style. We further create a *"User"* style for all scenes by simulating specific characters from TV series using their role profiles and dialogues (Wang et al., 2023a). We select five diverse characters from the SummScreen dataset (Chen et al., 2022a) to generate instructions with unique speaking styles, including a generalized child-speaking style due to the absence of actual child characters. We provide a comparison of different instructions in the appendix.

### 3.3.3 SPLIT AND STATISTICS

Since our focus is on scene adaptation after the training phase, we include only evaluation splits and use the training split of R2R for the GSA-R2R dataset. Given the two general building types (residential and non-residential) and three types of instructions, we design five splits for both validation and testing, with their details in the appendix. The splits are named using the format "Val/Test-R/N-Basic/Scene/User", where "R" denotes residential and "N" represents non-residential scenes. We exclude the val seen split, as adapting to a previously trained environment is unnecessary; thus, all environments in GSA-R2R are considered unseen. We compare GSA-R2R with other embodied navigation datasets in Tab. 1, demonstrating that GSA-R2R has the greatest number and diversity of scenes, paths, and instructions. Notably, we are the first to incorporate various speaking styles to address OOD instructions, as evidenced by the highest proportion of unseen vocabulary size.

### 3.3.4 DATASET EVALUATION

**Reliability.** Since our goal is to generate instructions with distinct styles and OOD characteristics, which intentionally diverge from the training data, traditional linguistic metrics such as BLEU (Papineni et al., 2002) or ROUGE (Lin, 2004) are not appropriate for evaluation. These metrics assess word-level alignment with ground truth instructions, whereas our instructions should align with the path observations. However, there is currently no automatic

---

[2]We include the evaluation splits in the training of EnvDrop to improve quality.

method to evaluate this alignment. Therefore, we conducted a user study to invite 15 participants to evaluate 20 randomly selected instructions from GSA-R2R. They judged whether the instructions accurately described the path and whether they exhibited a distinct speaking style. The results in Tab. 2 demonstrate around 80% alignment and a high proportion of distinct speaking styles, proving the reliability of our data. User instructions are judged as less styled, likely because their word-level changes are less noticeable when viewed individually.

**Diversity.** We provide a t-SNE analysis to compare the instructions from R2R and GSA-R2R, as shown in Fig. 4. Specifically, we use BERT (Devlin, 2018) to obtain embeddings for all instructions from these two datasets and then use t-SNE (Van der Maaten & Hinton, 2008) to project them into a 2D space for visualization. For clarity, we only display the results for the User Sheldon here; the results for other characters are presented in the appendix. The left image of Fig. 4 shows that the evaluation instructions in R2R are all ID data, sharing the same distribution as the training instructions. In contrast, our Scene and User instructions exhibit a distinctly different distribution from the training data, demonstrating the diversity of our dataset. Since the Basic instructions consist of speaker-generated and LLM-refined components, their cluster overlaps with the training cluster, which we still consider as ID data.

Table 2: Human evaluation of various GSA-R2R instructions. *Matching*: percentage of instructions that accurately describe the path. *Style*: percentage of instructions with a clear speaking style.

| Type | Matching↑ | Style↑ |
|---|---|---|
| EnvDrop | 52.2 | 13.0 |
| Basic | 80.0 | 14.7 |
| Scene | 76.6 | **96.1** |
| User | **83.3** | 57.6 |

## 4 EXPERIMENTS

### 4.1 GR-DUET

Although TourHAMT incorporates history information as additional input, its performance degrades for two reasons. First, representing each step with a single history embedding fails to capture the necessary spatial correlations for modeling visited nodes, especially when the history is extensive. Second, it only fine-tunes the model with new history embedding compositions, leading to a significant input distribution shift between pretraining and fine-tuning. To address these issues, we propose a novel memory-based method, graph-retained dual-scale graph transformer (GR-DUET), where the extended history embeddings are replaced by a global topological graph that retains information across episodes, ensuring comprehensive awareness of visited nodes.

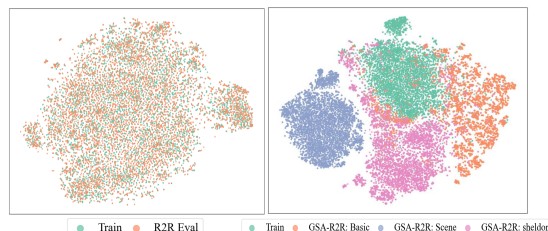

Figure 4: The t-SNE analysis of instructions from R2R (left) and GSA-R2R (right). Our instructions demonstrate significantly greater diversity compared to R2R and include OOD data.

Specifically, during inference, instead of maintaining separate graphs $\{\mathcal{G}_1, \mathcal{G}_2, \cdots, \mathcal{G}_m\}$ for each episode $\{EP_1, EP_2, \cdots, EP_m\}$, our agent maintains a single global graph $\mathcal{G}_g$ to continuously update the topological map while preserving observations at each node throughout the evaluation. At the start $t = 0$ of episode $k$, we utilize the data from the memory bank $\mathcal{M}_E$ to build the topological graph with all previously visited nodes $\mathcal{G}_g = \{\mathcal{G}_1, \mathcal{G}_2, \cdots, \mathcal{G}_{k-1}\}$. Each node represents a visited location, with node information including positions $\{x, y, z\}$ for spatial alignment and the visual observations $\mathbf{O}$ at that node. We reset the visited status of all nodes at the beginning of each episode to enable agents to only choose unvisited nodes as the next step for efficiency. By utilizing this global graph across episodes, GR-DUET can more effectively leverage historical information, enabling a deeper understanding of the environment and facilitating longer-term action planning, particularly after executing numerous instructions.

However, directly applying this mechanism in evaluation leads to the same distribution shift issue. Therefore, we modify the pretraining and fine-tuning stages to align the input distribution between training and inference. During pretraining, we provide the model with the complete ground truth

Table 3: Navigation performance of different VLN models in the test splits of R2R and GSA-R2R. †: Data leakage exists since ScaleVLN uses HM3D as training data.

| Methods | Test-R-Basic | | Test-N-Basic | | Test-R-User | | Test-N-User | | Test-N-Scene | | R2R-Test | |
|---|---|---|---|---|---|---|---|---|---|---|---|---|
| | SR↑ | SPL↑ | SR↑ | SPL↑ | SR↑ | SPL↑ | SR↑ | SPL↑ | SR↑ | SPL↑ | SR↑ | SPL↑ |
| HAMT (Chen et al., 2021) | 48 | 44 | 42 | 38 | 46 | 42 | 38 | 34 | 34 | 30 | 65 | 60 |
| DUET (Chen et al., 2022c) | 58 | 47 | 48 | 37 | 55 | 45 | 44 | 34 | 40 | 30 | 69 | 59 |
| BEVBert (An et al., 2023) | 58 | 45 | 46 | 35 | 55 | 43 | 44 | 33 | 39 | 27 | 73 | 62 |
| ScaleVLN (Wang et al., 2023b)† | 79 | 67 | 71 | 58 | 71 | 59 | 61 | 48 | 55 | 43 | 77 | 68 |
| NavGPT-2 (Zhou et al., 2024) | 58 | 45 | 48 | 35 | 56 | 45 | 45 | 33 | 44 | 58 | 72 | 60 |

topological map of the current environment to enable them to utilize the most abundant information. For fine-tuning, we adopt an environment-specific training strategy to assign individual graphs to each training environment. We set a threshold $\alpha$ as the maximum number of episodes to be included in the graph. Once $\alpha$ is reached, the graph is re-initialized to avoid constant prediction under a fully populated graph. To enhance graph coverage and path diversity, we further incorporate the PREVALENT dataset (Hao et al., 2020) as augmented data.

## 4.2 EXPERIMENTAL SETUP

**Baseline Methods**   We include two types of baseline methods. One is adaptation-based methods, including **TENT** (Wang et al., 2021), **SAR** (Niu et al., 2023), Back-Translation (**BT**), and proxy tasks (**MLM** (Devlin, 2018) & **MRC** (Lu et al., 2019)). The other is memory-based methods, including TourHAMT (Krantz et al., 2023) and OVER-NAV (Zhao et al., 2024). More details are provided in the appendix.

**Evaluation Metrics.**   We use the following metrics to evaluate the navigation performance: (1) Trajectory Length (**TL**): the total navigation distance in meters; (2) Navigation Error (**NE**): the distance between the stop location and the target; (3) Success Rate (**SR**): the ratio of agents stopping within 3 meters of the target; (4) Success rate weighted by Path Length (**SPL**) (Anderson et al., 2018a): SR normalized by the ratio between the shortest path length and the predicted path length. (5) Normalized Dynamic Time Warping (**nDTW**) (Ilharco et al., 2019): a measure of instruction fidelity by computing the similarity between the reference path and the predicted path.

**Implementation Details.**   We use GPT-4o from OpenAI's official API as the LLM for generating Scene and User instructions. All prompt templates are provided in the appendix. For baseline methods, we follow the implementation details in their official repositories, with two modifications. First, we use CLIP-ViT/B-16 (Radford et al., 2021) as the visual feature extractor for both the navigation and speaker models for fair comparison. Second, all models are evaluated using a batch size of 1 in an online manner during evaluation. In GR-DUET, we set the maximum number of episodes $\alpha = 50$. The best model is selected based on the average SPL across all validation splits. For each adaptation method, we conduct the evaluation three times with randomly sequenced instructions and report the mean and standard error for each metric.

## 4.3 MAIN RESULTS

In this section, we benchmark existing VLN methods and adaptation methods to show their performance for the GSA-VLN task.

### 4.3.1 HOW DO CURRENT VLN METHODS PERFORM IN GSA-R2R?

Various techniques have been incorporated into current VLN methods to achieve human-like performance in R2R, demonstrating their strong reasoning capabilities. We evaluate these methods without adaptation techniques to determine whether they can maintain the same performance in our diverse environments and instructions, as shown in Tab. 3. Except for the data leakage in ScaleVLN [3], the performance of other baselines in GSA-R2R is significantly lower than their performance in R2R, highlighting the challenges of our task. When comparing different environments, agents are better at residential scenes than non-residential ones due to the biased distribution of the training data. For different instruction types, all models perform best on Basic instructions, followed by User instruc-

---

[3]ScaleVLN uses all buildings from HM3D as augmented data for agent training, which includes the environments in GSA-R2R and a portion of the Basic instructions.

Table 4: Comparison of different adaptation methods in GSA-R2R with basic instructions.

| Methods | Test-R-Basic | | | | | Test-N-Basic | | | | |
|---|---|---|---|---|---|---|---|---|---|---|
| | TL | NE↓ | SR↑ | SPL↑ | nDTW↑ | TL | NE↓ | SR↑ | SPL↑ | nDTW↑ |
| *Baseline* | | | | | | | | | | |
| DUET (Chen et al., 2022c) | 13.1 | 4.2 | 57.7 | 47.0 | 55.6 | 14.8 | 5.3 | 48.1 | 37.3 | 45.9 |
| *Optimization-Based Methods* | | | | | | | | | | |
| +MLM (Devlin, 2018) | 13.1 ±0.1 | 4.1 ±0.1 | 57.9 ±0.2 | 47.3 ±0.1 | 55.9 ±0.2 | 13.1 ±0.2 | 5.3 ±0.1 | 48.3 ±0.5 | 38.8 ±0.5 | 48.4 ±0.3 |
| +MRC (Lu et al., 2019) | 13.1 ±0.1 | 4.2 ±0.1 | 57.7 ±0.1 | 47.0 ±0.1 | 55.6 ±0.1 | 14.7 ±0.1 | 5.3 ±0.1 | 48.1 ±0.1 | 37.3 ±0.1 | 45.9 ±0.1 |
| +BT (Wang et al., 2020) | 8.0 ±0.1 | 3.8 ±0.1 | 61.3 ±0.6 | 57.7 ±0.3 | 70.1 ±0.5 | 7.9 ±0.0 | 5.2 ±0.1 | 49.5 ±0.8 | 46.0 ±0.8 | 59.4 ±0.9 |
| +TENT (Wang et al., 2021) | 14.6 ±0.0 | 4.2 ±0.0 | 57.2 ±0.4 | 44.2 ±0.4 | 52.9 ±0.1 | 16.2 ±0.1 | 5.4 ±0.1 | 46.5 ±0.4 | 33.7 ±0.2 | 42.6 ±0.3 |
| +SAR (Niu et al., 2023) | 13.8 ±0.8 | 4.0 ±0.1 | 57.6 ±0.2 | 44.6 ±0.2 | 53.0 ±0.2 | 16.5 ±0.0 | 5.4 ±0.0 | 44.6 ±1.5 | 31.5 ±1.6 | 40.6 ±1.3 |
| *Memory-Based Methods* | | | | | | | | | | |
| TourHAMT (Krantz et al., 2023) | 11.6 ±0.1 | 7.4 ±0.1 | 14.9 ±0.1 | 12.2 ±0.1 | 34.7 ±0.1 | 9.4 ±0.1 | 7.7 ±0.1 | 11.0 ±0.2 | 8.6 ±0.2 | 32.2 ±0.1 |
| OVER-NAV (Zhao et al., 2024) | 14.1 ±0.1 | 6.7 ±0.0 | 22.3 ±0.3 | 16.8 ±0.2 | 37.1 ±0.1 | 11.4 ±0.1 | 7.1 ±0.1 | 16.6 ±0.2 | 13.0 ±0.1 | 35.0 ±0.2 |
| GR-DUET (ours) | 9.4 ±0.0 | 3.1 ±0.0 | 69.3 ±0.2 | 64.3 ±0.1 | 71.4 ±0.1 | 8.9 ±0.0 | 4.4 ±0.0 | 56.6 ±0.1 | 51.5 ±0.1 | 61.0 ±0.1 |

Table 5: Comparison of different adaptation methods in GSA-R2R with User instructions.

| Methods | Child | | Keith | | Moira | | Rachel | | Sheldon | |
|---|---|---|---|---|---|---|---|---|---|---|
| | SR↑ | SPL↑ | SR↑ | SPL↑ | SR↑ | SPL↑ | SR↑ | SPL↑ | SR↑ | SPL↑ |
| *Baseline* | | | | | | | | | | |
| DUET | 54.3 | 44.1 | 56.0 | 46.3 | 52.3 | 43.3 | 56.3 | 46.4 | 54.0 | 44.4 |
| *Optimization-Based Methods* | | | | | | | | | | |
| +MLM | 54.5 ±0.2 | 44.7 ±0.2 | 56.4 ±0.3 | 46.8 ±0.3 | 53.8 ±0.3 | 43.6 ±0.4 | 56.8 ±0.5 | 46.6 ±0.6 | 54.5 ±0.4 | 44.2 ±0.3 |
| +MRC | 54.4 ±0.2 | 44.2 ±0.1 | 56.0 ±0.1 | 46.3 ±0.1 | 52.3 ±0.2 | 43.3 ±0.1 | 56.0 ±0.1 | 46.2 ±0.2 | 53.7 ±0.2 | 44.2 ±0.4 |
| +BT | 57.5 ±0.7 | 54.0 ±0.9 | 61.2 ±0.3 | 57.9 ±0.1 | 57.3 ±0.5 | 54.0 ±0.6 | 61.6 ±0.8 | 58.1 ±0.7 | 57.6 ±0.5 | 54.3 ±0.5 |
| +TENT | 54.3 ±0.2 | 41.7 ±0.1 | 55.4 ±0.2 | 43.8 ±0.2 | 51.7 ±0.2 | 41.0 ±0.1 | 55.0 ±0.2 | 43.2 ±0.2 | 53.0 ±0.2 | 41.9 ±0.1 |
| +SAR | 54.5 ±0.5 | 41.5 ±0.4 | 54.9 ±0.3 | 43.1 ±0.2 | 51.0 ±0.4 | 40.3 ±0.6 | 55.3 ±0.5 | 43.0 ±0.6 | 52.9 ±0.2 | 41.4 ±0.4 |
| *Memory-Based Methods* | | | | | | | | | | |
| TourHAMT | 14.6 ±0.2 | 12.0 ±0.2 | 15.1 ±0.2 | 12.3 ±0.1 | 13.9 ±0.1 | 11.3 ±0.1 | 15.3 ±0.1 | 12.5 ±0.1 | 14.4 ±0.1 | 11.8 ±0.1 |
| OVER-NAV | 20.9 ±0.1 | 16.1 ±0.2 | 20.5 ±0.1 | 16.4 ±0.1 | 19.5 ±0.2 | 15.4 ±0.2 | 20.6 ±0.3 | 16.2 ±0.2 | 20.5 ±0.1 | 16.2 ±0.1 |
| GR-DUET (ours) | 65.2 ±0.1 | 59.7 ±0.1 | 66.7 ±0.1 | 62.0 ±0.1 | 60.9 ±0.2 | 56.2 ±0.2 | 67.1 ±0.1 | 62.2 ±0.1 | 63.9 ±0.1 | 58.9 ±0.1 |

tions, and finally Scene instructions. This aligns with the cluster distance from the training data observed in Fig. 4, emphasizing the importance of studying the OOD problem in VLN.

### 4.3.2 HOW DO ADAPTATION METHODS PERFORM IN GSA-R2R?

**Environment Adaptation.** We first test the adaptation methods with different environments using Basic instructions, as shown in Tab. 4. Among the optimization-based methods, TTA techniques like TENT and SAR perform worse than vanilla DUET. This is because entropy-based methods assume a positive correlation between confidence and accuracy. However, in sequential decision-making processes like VLN, errors accumulate over time, making entropy measures meaningless after an incorrect step. The Back-Translation (BT) method shows improvement as the Basic instructions

Table 6: Comparison of different adaptation methods in GSA-R2R with Scene instructions.

| Methods | Test-N-Scene | | | | |
|---|---|---|---|---|---|
| | TL | NE↓ | SR↑ | SPL↑ | nDTW↑ |
| *Baseline* | | | | | |
| DUET | 14.9 | 6.4 | 39.6 | 30.1 | 40.9 |
| *Optimization-Based Methods* | | | | | |
| +MLM | 14.3 ±0.1 | 6.5 ±0.1 | 39.8 ±0.1 | 30.5 ±0.1 | 41.1 ±0.1 |
| +MRC | 14.9 ±0.1 | 6.4 ±0.1 | 39.7 ±0.1 | 30.2 ±0.1 | 40.9 ±0.1 |
| +BT | 8.4 ±0.0 | 6.3 ±0.2 | 41.2 ±1.5 | 38.2 ±1.2 | 51.3 ±1.2 |
| +TENT | 16.4 ±0.1 | 6.3 ±0.1 | 40.6 ±0.2 | 28.9 ±0.2 | 38.9 ±0.2 |
| +SAR | 16.3 ±0.5 | 6.0 ±0.2 | 41.4 ±0.6 | 29.1 ±0.3 | 39.0 ±0.3 |
| *Memory-Based Methods* | | | | | |
| TourHAMT | 7.3 ±0.1 | 8.1 ±0.1 | 9.7 ±0.1 | 8.0 ±0.1 | 32.3 ±0.1 |
| OVER-NAV | 11.8 ±0.1 | 7.6 ±0.2 | 16.7 ±0.4 | 12.6 ±0.2 | 34.6 ±0.3 |
| GR-DUET (ours) | 10.1 ±0.0 | 5.5 ±0.0 | 48.1 ±0.1 | 42.8 ±0.1 | 53.7 ±0.1 |

closely resemble the authentic data from the speaker. For proxy tasks, MRC performs similarly to DUET, as this coarse-grained task is not beneficial once the agent already has robust representations through extensive training, which is also observed in ScaleVLN (Wang et al., 2023b). MLM provides marginal improvement. While it helps learn better textual features, it is not optimized together with action prediction, preventing the model from leveraging these improved language features. For memory-based methods, both TourHAMT and OVER-NAV performed significantly worse than even vanilla HAMT. This is due to the larger number of instructions in GSA-R2R compared to their training data, resulting in excessively long history embeddings as input, which confuses the model. In contrast, GR-DUET achieves the best performance in both Residential (R) and Non-residential (N) splits, with an 11.6% and 8.5% SR increase, respectively. This demonstrates its effectiveness in helping agents adapt to both ID and OOD environments.

**Instruction Adaptation.** We also evaluate these methods across different instruction styles. Tab. 5 shows the results for User instructions from the five characters, while Tab. 6 presents their performance on Scene instructions. The performance of DUET suggests that different speaking styles

Table 7: Ablation study on the pretraining and augmented data in GR-DUET.

| Pretrain | Aug. | Test-R-Basic | | Test-N-Basic | | Test-N-Scene | |
|---|---|---|---|---|---|---|---|
| | | SR↑ | SPL↑ | SR↑ | SPL↑ | SR↑ | SPL↑ |
| × | × | 56.8 ±0.1 | 47.5 ±0.1 | 45.4 ±0.1 | 35.3 ±0.1 | 38.2 ±0.0 | 28.9 ±0.0 |
| × | ✓ | 54.0 ±0.1 | 41.4 ±0.1 | 43.0 ±0.0 | 30.9 ±0.1 | 35.9 ±0.4 | 27.4 ±0.1 |
| ✓ | × | 59.9 ±0.1 | 48.2 ±0.1 | 47.9 ±0.1 | 35.3 ±0.1 | 43.7 ±0.2 | 33.6 ±0.0 |
| ✓ | ✓ | **69.3** ±0.2 | **64.3** ±0.1 | **56.6** ±0.1 | **51.5** ±0.1 | **48.1** ±0.1 | **42.8** ±0.1 |

Table 8: Ablation study on different graph construction mechanisms in GR-DUET.

| Method | $\alpha$ | Test-R-Basic | | Test-N-Basic | | Test-N-Scene | |
|---|---|---|---|---|---|---|---|
| | | SR↑ | SPL↑ | SR↑ | SPL↑ | SR↑ | SPL↑ |
| Proportion | 0.25 | 61.2 ±0.1 | 53.3 ±0.1 | 50.9 ±0.1 | 39.2 ±0.1 | 41.4 ±0.2 | 30.6 ±0.1 |
| | 0.50 | 64.8 ±0.1 | 56.8 ±0.1 | 50.8 ±0.1 | 40.4 ±0.1 | 37.8 ±0.0 | 29.4 ±0.0 |
| | 0.75 | 57.7 ±0.1 | 48.7 ±0.0 | 48.3 ±0.2 | 37.5 ±0.1 | 40.7 ±0.1 | 28.7 ±0.1 |
| | 1.00 | 66.2 ±0.3 | 58.5 ±0.2 | 55.7 ±0.2 | 46.0 ±0.1 | 47.5 ±0.2 | 40.9 ±0.1 |
| Buffer | 1 | 57.6 ±0.1 | 35.1 ±0.1 | 45.1 ±0.1 | 24.9 ±0.0 | 36.9 ±0.1 | 20.6 ±0.1 |
| | 50 | 69.3 ±0.2 | **64.3** ±0.1 | **56.6** ±0.1 | **51.5** ±0.1 | **48.1** ±0.1 | **42.8** ±0.1 |
| | 100 | **69.7** ±0.1 | 63.2 ±0.1 | 56.1 ±0.1 | 48.5 ±0.1 | 47.9 ±0.2 | 37.1 ±0.1 |
| | 150 | 67.5 ±0.0 | 59.3 ±0.0 | 54.8 ±0.2 | 44.6 ±0.1 | 46.2 ±0.1 | 37.9 ±0.1 |

introduce varying levels of difficulty for VLN models in interpreting instructions. Most results are consistent with those from the environment adaptation results, with two notable exceptions. First, TTA methods achieve 1% SR increase in Scene instructions, but this advantage disappears in User instructions. We attribute this to their different language patterns. Scene instructions tend to include conversational fillers, which is a noticeable pattern that optimization-based methods can capture. In contrast, User instructions focus on word variations, which is much more challenging. Second, the improvement from Back-Translation diminishes due to the domain shift between the authentic and evaluation instructions, indicating that this method is effective for environment adaptations but not for instruction adaptations. Lastly, Our GR-DUET again achieves significant performance improvements with a maximum SR increase of 11% across all splits, demonstrating its general applicability to both types of adaptations.

## 4.4 ABLATION STUDY

We conduct two ablation studies on our GR-DUET. Tab. 7 proves that only adding PREVALENT is detrimental due to the introduced instruction noises and only incorporating full graphs during pretraining is limited by the lack of path diversity during fine-tuning. Combining both approaches leads to significant improvements, demonstrating the effectiveness of this strategy. In Tab. 8, we experiment with two strategies to simulate the graph construction process in fine-tuning. The "proportion" method randomly provides a specific proportion of ground-truth graphs to agents while the "memory" method is the realization in GR-DUET with a maximum capacity for memorizing episodes. For simplicity, we use the same symbol, $\alpha$, to represent both the proportion and buffer size. The results prove that memory methods outperform proportion methods, as they more closely align with the gradual expansion of the global graph during inference. For memory methods, performance initially improves with increasing buffer size, then declines. This trend is expected as a small buffer cannot cover graphs adequately, while an excessively large buffer leads to inefficiencies.

## 5 CONCLUSION

In this paper, we introduce the GSA-VLN task to highlight the challenges faced by VLN agents operating in persistent environments, where long-term memory and model updates are required to adapt to specific settings. To thoroughly evaluate agent adaptability, we create the GSA-R2R dataset, which significantly expands the quantity and diversity of environments and instructions, including both ID and OOD data for evaluation. We benchmark popular VLN models and adaptation methods on GSA-R2R and propose GR-DUET, a novel model that integrates global graphs with an environment-specific training strategy, achieving state-of-the-art results. In the future, we aim to explore more unsupervised learning approaches to further enhance agent performance in GSA-R2R.

ACKNOWLEDGMENTS

This work is supported by projects DP230101753 funded by the Australian Research Council, and CSIRO's Science Leader project R-91559. This work is also supported by the Centre for Augmented Reasoning, an initiative by the Department of Education, Australian Government.

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

## A  APPENDIX

This document provides additional method details, supplementary experiments, and further analysis to complement the main paper, including:

- Appendix A.1: detailed descriptions of the baseline methods.
- Appendix A.2: more explanations of the process for generating the GSA-R2R dataset.
- Appendix A.3: additional statistics and visualizations of the GSA-R2R dataset.
- Appendix A.4: discussion of the feasibility of deploying GR-DUET in real-time systems.
- Appendix A.5: detailed analysis of the scalability of GR-DUET.
- Appendix A.6: comparison with LLM-based VLN methods.
- Appendix A.7: quantitative analysis of the adaptation speed of GR-DUET.
- Appendix A.8: more details about the human study.
- Appendix A.9: Detailed justification for selecting five characters for User instructions.
- Appendix A.10: prompt templates used for generating GSA-R2R.
- Appendix A.11: discussions on the limitations and future directions of this work.

### A.1  BASELINE METHODS

In this section, we provide detailed descriptions of the baseline adaptation methods used in our experiments, which are categorized into two types: optimization-based methods, which update model parameters during evaluation, and memory-based methods, which use data from the memory bank as additional input.

#### A.1.1  DUET DETAILS

We adopt DUET (Chen et al., 2022c) as our baseline model, which is an enhanced version of HAMT (Chen et al., 2021). The HAMT model employs a transformer-based network to encode instructions, visual observations, and navigation history. These components are first processed by individual encoders and subsequently integrated through a cross-modal transformer (Vaswani, 2017). Building on the foundation of HAMT, DUET introduces a real-time topological map to track visited nodes and leverages graph transformers to enable global action decisions. Unlike HAMT, which is restricted to selecting actions only from neighboring nodes of the current location—leading to navigation inefficiencies—DUET expands its action space to include all nodes in the topological map. This allows DUET to efficiently navigate to distant nodes using a path planner, significantly improving navigation efficiency. Moreover, DUET employs a dual-level encoding architecture to encode both fine-grained features of visual observations and coarse-grained features of the topological map. These representations are fused with instructions to capture cross-modal relationships, facilitating more effective action predictions. For agent training, DUET adopts the two-stage training paradigm introduced by HAMT, consisting of pretraining on proxy tasks and fine-tuning on downstream tasks. During fine-tuning, DUET employs a pseudo-interactive demonstrator to enhance exploration, thereby improving generalization performance.

#### A.1.2  OPTIMIZATION-BASED ADAPTATION METHODS

Due to the lack of ground-truth paths, optimization-based adaptation methods rely on unsupervised and self-supervised training strategies to adapt the model to specific environments.

**Entropy-Based Methods.**  Online Test-Time Adaptation (TTA) methods (Liang et al., 2024) focus on minimizing the entropy of agent predictions to make agents more confident in their decisions by exploiting the positive correlation between entropy and errors. Since the VLN task is formulated as a classification task on neighboring viewpoints, these methods are directly applicable to the GSA-VLN task. We employ two widely used methods in this field, TENT (Wang et al., 2021) and SAR (Niu et al., 2023), to evaluate whether these TTA techniques can help agents adapt to specific environments and speaking styles. TENT takes entropy minimization as an optimization objective

on all test samples while SAR further takes the sharpness of entropy into consideration to only consider those reliable test samples instead of all samples to adapt to more diverse settings.

**Back-Translation.**    If we consider the visited nodes in previous episodes as exploring the environment, the scene adaptation problem in the current episode can be regarded as the pre-explore setting (Wang et al., 2019) in the standard VLN task. A key method in this setting is back-translation (Wang et al., 2020), which involves using a trained speaker to generate instructions for randomly sampled paths consisting of visited nodes, followed by teach-forcing imitation learning on this synthetic data. While this method has proven effective in the pre-explore setting of R2R, its performance for scene adaptation in GSA-R2R is less promising. This is primarily due to the introduction of diverse instruction styles, which causes the synthetic instruction to diverge significantly from the actual evaluation instructions. Another key difference is that traditional back-translation operates on the full navigation graph, while in GSA-VLN, it is limited to the current constructed graph, which is only part of the full graph. This leads to two issues. First, the sampled paths may not always be the shortest between two viewpoints since the local optimal way may differ from the global one. Second, with the generated instruction-trajectory pairs, the agent performs pure imitation learning without exploration, as the entire process occurs within the agent's internal model, not in the simulator, and the memory bank lacks data from unvisited nodes.

**Proxy Tasks.**    One important approach for unsupervised training in VLN is using proxy tasks to update the model, which is widely employed during the pretraining stage of VLN models. We employ two common proxy tasks, Masked Language Modelling (MLM) (Devlin, 2018) and Masked Region Classification (MRC) (Lu et al., 2019), to individually assess whether they can help agents adapt to specific visual environments or distinct instruction styles. MLM randomly masks 15% of instruction tokens and utilizes the output embeddings to predict the masked tokens. MRC similarly masks 15% of visual observations and uses the output embeddings to predict the semantic labels of the masked regions where the label is obtained from an image classification model (Dosovitskiy, 2020) pretrained on ImageNet (Deng et al., 2009). Since both MLM and MRC are used in the pretraining stage of DUET, we directly reuse their prediction heads in our adaptation process.

### A.1.3    MEMORY-BASED ADAPTATION METHODS

The memory-based adaptation methods explicitly maintain information from previous episodes as additional input to help decision-making without updating model parameters. We evaluate two IVLN methods as representatives of this category: TourHAMT (Krantz et al., 2023) and OVER-NAV (Zhao et al., 2024). TourHAMT incorporates information from previous episodes as additional history embeddings for HAMT to enable reasoning across episodes. OVER-NAV performs real-time detection of keywords from instructions within observations and stores the detection results in an Omnigraph, which describes the object distribution to facilitate navigation.

### A.2    DATA GENERATION IN GSA-R2R

Here, we provide additional details on our instruction orchestration pipeline. In the first stage, we employ the EnvDrop speaker (Tan et al., 2019) trained on R2R to generate initial instructions for each sampled path. Unlike previous works, we include data from the validation splits in the training process of the speaker to enhance instruction quality, as these results are used for evaluation rather than training. We selected EnvDrop over recent speakers because of its widespread use and established performance in VLN tasks. However, the generated instructions still contain noise and inaccuracies that may not align with the path.

To address this, the second stage involves model-based selection, which uses a navigation model to identify incorrect instructions. Specifically, we use unselected paths from the 150 environments in GSA-R2R to train a DUET (Chen et al., 2022c) model, which is then evaluated on the selected instructions to determine whether the trained agent can successfully execute the instructions, serving as an indicator of instruction feasibility. For environments with a limited number of unselected paths, we apply two-fold cross-validation to finetune the DUET on the specific environment using all available paths. The trained DUET achieves a 73.6% SR for instructions generated on the selected paths. For instructions that fail, we leverage the multi-modal reasoning capability of GPT-4o to identify misalignments between the trajectory and instruction, providing corrected versions using

specifically designed path visualization prompts. These prompts consist of two parts: a sequence of panoramas from the viewpoints and a top-down view of the path on the map. Various methods were tested for presenting the panoramas, as it remains challenging for the language model to fully comprehend the spatial relationships between panoramas. We finally chose to display the first-person view facing the direction of travel, with red arrows indicating the intended path, with examples in Appendix A.3.3. Additionally, we employ the Chain-of-Thought (CoT) technique (Wei et al., 2022) to require the LLM to detect any issues in the speaker-generated instructions, list the errors, and finally provide a corrected version with an explanation of the modifications. We refer to these instructions, which are either good or refined, as *"Basic"* instructions in GSA-R2R, denoting their concise and style-neutral nature commonly found in VLN datasets.

Building on the *Basic* instructions, the third stage rephrases them to reflect distinct speaking styles. Since real-world instructors can be specific individuals, such as homeowners or specialized users of a building type, we introduce two speaking styles: *"Scene"* and *"User"*. For non-residential buildings, We use an LLM to identify potential users based on the building type, then randomly select one to serve as the instructor to rephrase each instruction. We make the LLM align with the speaking style of the selected speaker while keeping the directional information unchanged, thus generating the Scene instructions.

For the *User* instructions, inspired by recent studies on role-playing capabilities in language models (Chen et al., 2024b), we use GPT-4o to simulate specific characters and rephrase the *Basic* instructions for both residential and non-residential scenarios. Current role-playing studies (Chen et al., 2024b) include two types: persona-based, which relies on persona descriptions, and character-based, which mimics specific characters' behaviors. We found that persona-based methods often include irrelevant words, making the instructions less realistic. For example, a persona described as a *"bookshop owner providing reading material from various historical periods"* resulted in metaphors like *"Stroll beyond the bed, akin to stepping through the pages of a medieval manuscript"*, which are inappropriate for navigation tasks. Therefore, we adopted the character-based method.

Specifically, we first identify characters with distinct speaking styles from TV series with mainly daily scene dialogue in the SummScreen dataset (Chen et al., 2022a). Following RoleLLM (Wang et al., 2023a), we build a role profile and retrieve the top-5 relevant dialogue for each character from the scripts as context for prompting. We then generate rephrased instructions using the same five Basic instructions for each character and calculate word overlap to identify which character has the most distinct style. Considering the diversity in speaking styles, age, and gender, we selected five fictional characters to generate the *User* instructions: Rachel from *FRIENDS*, Moira from *Schitt's Creek*, Keith from *Veronica Mars*, and Sheldon from *The Big Bang Theory*. Since there were no child characters in these TV series, we simulated a general child-speaking style rather than that of a specific character.

Two notable features emerged in the rephrased instructions. First, the vocabulary size expanded to include more uncommon words, such as *"sashaying"* and *"meander"*. Second, additional non-navigation words, such as conversational fillers and character catchphrases, were incorporated. The comparison of different instructions is illustrated in Fig. 5. Adapting agents to these varied speaking patterns is beneficial for improving performance in the GSA-VLN task. We include all the prompt templates used in Appendix A.10.

## A.3 DATASET STATISTICS

In this part, we provide more statistics and visualizations of our GSA-R2R dataset to provide a more comprehensive evaluation.

### A.3.1 LINGUISTIC METRICS.

As stated in the main paper, our goal is to generate diverse instructions, making traditional linguistic metrics less suitable, as they compare generated instructions with ground truth instructions rather than with the path itself. Therefore, we provide linguistic metrics here for reference only.

**Basic Instructions.** To demonstrate the effectiveness of the second stage of our instruction orchestration pipeline which utilizes the VLM for instruction refinement, we apply the same method to the paths from the validation unseen split of R2R, allowing for a comparison with ground truth

| Train | R2R | Basic 📋 | Scene 🏠 | User 🧑‍🤝‍🧑 |
|---|---|---|---|---|
| Walk down one flight of stairs and stop on the landing. | Walk up stairs. Wait at top of stair landing. | Exit the room and turn right. walk down the hallway and stop in front of the two white chairs. | Alright, so what you're gonna do is walk straight down the hallway, okay? Then, you'll make a left turn. Keep going until you see the office, and once you're inside, you'll turn right again. Finally, walk into the office and stop right there. Got it? *(School)* | Okay, so first, you gotta walk by the place where grown-ups drink stuff. Keep going straight, don't turn! Then, go into the room right in front of you. There'll be a shiny, marble room with rock benches. Walk all the way to the big water thingy and stop there! *(Child)* |
| Leave the room, and take a left. Take a right at the end of the hall, and take the first right. Enter the room, and stop in the doorway. | Turn slight left and walk down the aisle to the back. Turn right then go around the column. Stop near the poster in the window. | Walk through the door and turn right. walk into the first door on the right and stop. | So, just turn around and head out of the room, okay? Then, take a left and stroll down the hallway. You'll want to keep going past that glass door, and then you'll see the room with the white chair. That's where you'll want to go. *(Salon)* | Traverse the hallway and descend the staircase. Halt at the terminus of the stairs. *(Sheldon)* |
| Go up the stairs, and go right. Veer right, and enter the suite on the right. Stop once you are in the doorway. | Go out the kitchen using the opening on the left then turn right and head toward the stairs and wait there. | Walk through the doorway and turn left. walk into the room on the left and stop in front of the bathtub. | Alright, listen up! First, you're gonna walk straight ahead, okay? Then, when you see the mirrors on your left, make a left turn. Keep moving straight, and when you hit the row of treadmills, take a right turn. Keep going straight again until you reach the stretching area, and stop right there. Got it? *(Gym)* | Stroll past the bar, continue onward, and enter the room straight ahead. You'll find yourself in a grand marble chamber with stone benches. Stand majestically before the water basin. *(Moira)* |
| Turn right and walk through the kitchen. Then turn left and enter the foyer. Wait at the bottom of the stairs. | Turn right and walk toward the stairs. Turn right and wait near the lamp before you get to the stairs. | Walk out of the laundry room and turn left. walk down the hall and turn left. walk into the living room and stop. | So, when you come out of the bathroom, just go ahead and take a left. Then, you'll want to stroll down the hallway a bit and make another left at the very first door you see. Keep going past the couch there, and then just stop right in the doorway. *(Hotel)* | Stroll outta the bedroom, swing a left, step outside, and just chill. *(Rachel)* |
| Turn around and walk down the hallway. Turn right and walk up the small set of steps. Wait there. | Walk all the way up the staircase, and turn left at the top of the stairs, and enter the very first room with a visible white bed, and stop there. | Turn left and walk down the hallway. turn left and walk past the kitchen. walk past the kitchen and stop in the doorway to the kitchen. | So, um, just head past the kitchen, okay? Then, you're gonna want to make a left. Keep going past the kitchen and take that left turn. After that, you'll see the counter. Just walk right by it and stop in front of the counter. Got it? *(Shop)* | Spin around, walk out of the room, take a left, stroll down the hallway, take another left past the dining room table, and wait in the kitchen doorway. *(Keith)* |

Figure 5: Comparison of instructions between R2R and various speaking styles in GSA-R2R. Words that represent the speaking style are underlined. Our instructions demonstrate significantly greater diversity and distinctiveness in speaking styles.

Table 9: Linguistic evaluations of different instruction generation methods on val unseen split of R2R.

| Methods | SPICE↑ | BLEU-1↑ | BLEU-4↑ | Meteor↑ | Rouge↑ |
|---|---|---|---|---|---|
| BT-Speaker (Fried et al., 2018) | 18.8 | 52.2 | 14.2 | 22.8 | 34.6 |
| EnvDrop (Tan et al., 2019) | 18.1 | 68.4 | 23.7 | 22.5 | **45.8** |
| Ours | **21.4** | **69.9** | **24.0** | **23.2** | 45.3 |

instructions. The results are presented in Tab. 9. For most metrics, our method outperforms the original EnvDrop instructions, highlighting the effectiveness of our approach. However, for Rouge, which is based on word-level matching, the speakers with a fixed vocabulary perform better than our open-vocabulary approach. This outcome is reasonable but does not necessarily reflect the overall quality of the instructions.

**Scene and User Instructions.** We evaluate Scene and User instructions compared to the Basic instructions to show their differences in Tab. 10. The results provide two insights. First, User instructions are more similar to Basic instructions compared to Scene instructions. This is likely due to the rephrasing techniques used, where Scene instructions include more conversational fillers, leading to greater divergence from the Basic style. Second, the five characters exhibit varying degrees of deviation from the Basic instructions. Although they are generated using the same method, the use of role profiles and dialogue history enables the LLM to capture distinct patterns in each character's speaking style, resulting in diverse instructions. This demonstrates the effectiveness of our approach in producing character-specific instructions.

**Navigation Evaluation.** We further apply a trained navigation model (DUET) to evaluate the generated instructions in a zero-shot manner, demonstrating the effectiveness of our refinement and rephrasing methods from another perspective. The results are shown in Tab. 11. Comparing the speaker-generated instructions in Stage 1 to the refined Basic instructions in Stage 2, we observe a performance increase, indicating that our VLM-based instruction refinement successfully corrects noisy instructions. After introducing diverse speaking styles in Stage 3, both Scene and User instructions show a performance drop, suggesting that the incorporation of different speaking styles introduces additional challenges for the navigation task.

Table 10: Evaluation of instructions of different speaking styles in GSA-R2R.

| Instructions | | SPICE↑ | BLEU-1↑ | BLEU-4↑ | Meteor↑ | Rouge↑ |
|---|---|---|---|---|---|---|
| Scene | | 37.6 | 39.5 | 16.2 | 31.8 | 51.1 |
| User | Sheldon | 44.1 | 61.5 | 33.3 | 32.1 | 63.0 |
| | Moira | 40.6 | 59.3 | 30.4 | 31.3 | 61.3 |
| | Rachel | 51.9 | 68.0 | 40.9 | 36.9 | 70.0 |
| | Keith | 48.4 | 61.3 | 33.6 | 33.7 | 65.0 |
| | Child | 42.2 | 49.4 | 21.3 | 35.7 | 57.6 |

Table 11: Navigation performance in different instructions on validation splits of GSA-R2R.

| Instructions | Val-N-Scene | | | Val-R-User | | | Val-N-User | | |
|---|---|---|---|---|---|---|---|---|---|
| | SR | SPL | nDTW | SR | SPL | nDTW | SR | SPL | nDTW |
| Stage 1 | 39.2 | 30.8 | 43.2 | 56.4 | 46.9 | 56.0 | 43.9 | 32.9 | 42.4 |
| +Stage 2 | 42.8 | 33.5 | 44.8 | 59.1 | 49.1 | 57.0 | 46.4 | 34.8 | 43.4 |
| +Stage 3 | 39.3 | 31.1 | 43.6 | 55.6 | 46.5 | 55.6 | 44.1 | 33.6 | 43.6 |

### A.3.2 DATASET SPLITS

In Tab. 12, we provide detailed information on the splits in GSA-R2R. Since we have two types of environments and three kinds of instructions, there are five splits for both validation and test sets, as residential houses do not have Scene instructions. Each split contains at least 10 buildings, with 600 instruction-trajectory pairs for each building. In the User splits, each path has five instructions corresponding to five different characters.

### A.3.3 DATASET VISUALIZATIONS

In this section, we present additional visualizations of the GSA-R2R dataset to further illustrate its characteristics.

**Instruction Length.** We show the statistics on instruction length for different instruction types from GSA-R2R on the left side of Fig. 6. These results support our observation about the different speaking patterns between Scene and User instructions. Scene instructions, which include conversational fillers, tend to be the longest, while User instructions make changes primarily at the word level, resulting in lengths similar to the Basic instructions.

Table 12: GSA-R2R splits: each cell shows the split name, the number of scans, and the number of instruction-trajectory pairs.

| Split | Validation | | Test | |
|---|---|---|---|---|
| | Residential | Non-residential | Residential | Non-residential |
| Basic | Val-R-Basic 15 scans 9K pairs | Val-N-Basic 10 scans 6K pairs | Test-R-Basic 24 scans 14.4K pairs | Test-N-Basic 15 scans 9K pairs |
| Scene | - | Val-N-Scene 10 scans 6K pairs | - | Test-N-Scene 15 scans 9K pairs |
| User | Val-R-User 15 scans 45K pairs | Val-N-User 10 scans 30K pairs | Test-R-User 21 scans 12.6K pairs | Test-N-User 15 scans 45K pairs |

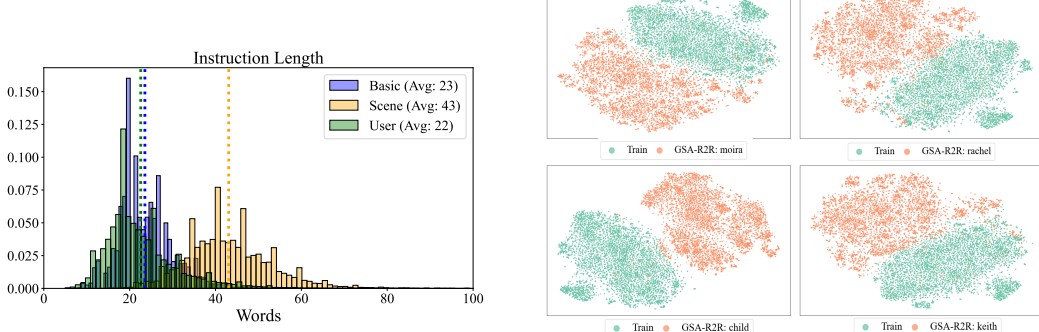

Figure 6: **Left**: The distributions of the instruction length for different types of instructions in GSA-R2R. **Right:** The t-SNE analysis of User instructions from other characters in GSA-R2R.

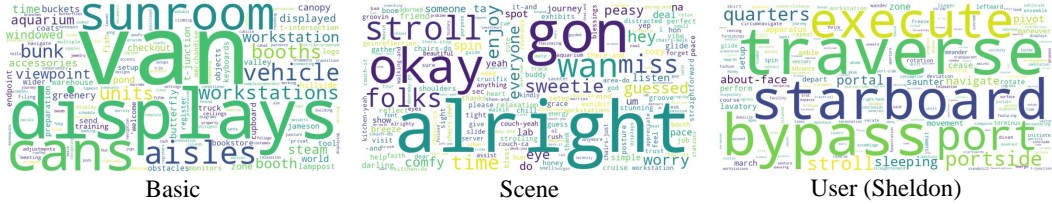

Figure 7: Word clouds of nouns and verbs absent from the training data across various speaking styles in the GSA-R2R dataset.

**t-SNE Visualization for Other Characters.** We present the t-SNE visualizations of other characters in the User instructions of GSA-R2R on the right side of Fig. 6. Each character's instructions form independent clusters, separate from the training distribution, confirming that our User instructions represent OOD data. We do not combine all users into a single figure, as they are based on the same Basic instructions, resulting in similar semantic meanings, which makes it difficult to distinguish between the clusters if visualized together.

**Word Cloud.** In Fig. 7, we present the word cloud visualization showing the nouns and verbs not found in the training data across the three types of instructions in GSA-R2R. The words in the Basic instructions are all from the refined results since the trained speaker uses the same vocabulary as the training data. Scene instructions include many conversational fillers such as *"Alright"* and *"Okay"*, while the User instructions of Sheldon include more complex and specialized terms like *"starboard"* and *"traverse"*. These results highlight the significant diversity in GSA-R2R instructions and the distinct differences between various speaking styles.

**Trajectory Visualization.** We provide qualitative examples of the environments, trajectories, and instructions from GSA-R2R in Fig. 8. The trajectories consist of a sequence of panoramas which are the observations from the start point to the destination. Each panorama is numbered in the upper left corner to indicate its sequence in the path. The direction each image faces shows the origin of movement, while red arrows mark the direction of travel. These visualizations are also used as prompts for the VLM to refine the instructions, as well as for participants in human evaluations. From these examples, it is clear that GSA-R2R incorporates a diverse range of environments and instructions and provides valuable resources for future VLN research.

## A.4 PRACTICAL DEPLOYMENT FEASIBILITY

In this section, we provide a detailed analysis of the computational and memory overhead associated with our GR-DUET to prove that our proposed method can be effectively deployed in real-world systems, such as robotics or autonomous agents. We use the largest environment in GSA-R2R as an example to demonstrate the resource requirements of GR-DUET during inference.

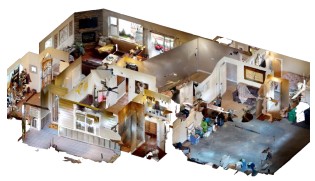
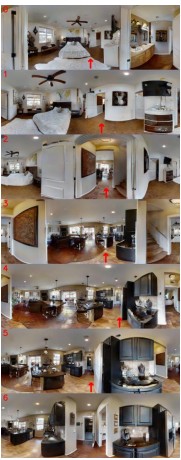
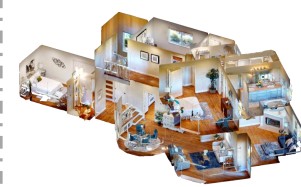
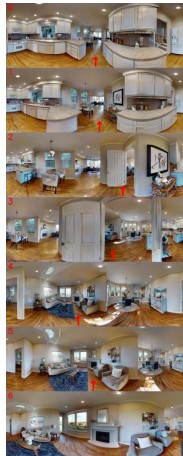

*Basic*: **walk straight down the hallway and turn right. walk past the kitchen and turn right. walk into the kitchen and stop.**

*User:* **Traverse straight through the culinary zone into the dining area, enter the hallway ahead, proceed through it, enter the living room, and halt at the fireplace.**

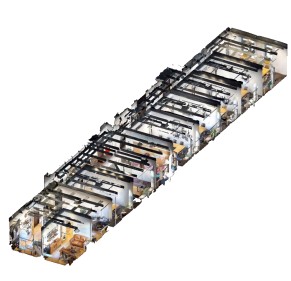
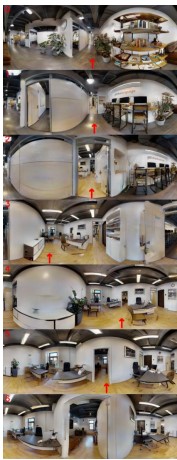

*Scene:* **Okay, so what you need to do is walk straight down the hallway, and then you'll make a right turn. Keep going straight past the desk and chairs-don't get distracted!-and then turn left just before you reach the door. Walk straight through the office area and head into the room that's directly in front of you. Once you get there, just stop.**

Figure 8: Qualitative examples of the environments, trajectories, and instructions from GSA-R2R.

**Memory Requirements**   GPU Memory: GR-DUET requires a peak of 4.3 GB of GPU memory during inference, which is well within the capacity of modern GPUs. For instance, it can be deployed on terminal servers equipped with hardware like the NVIDIA Jetson AGX Orin or similar devices. CPU Memory: The method requires at most 5.3 GB of RAM, which is easily manageable by most modern robotics platforms.

**Computational Overhead**   Inference Latency: GR-DUET achieves an average inference latency of 67 milliseconds per frame, allowing efficient navigation in most real-world environments. Throughput: The system processes 15 frames per second, which is sufficient for environments where navigation speed is moderate and does not require high-frequency updates, such as indoor environments with static obstacles.

**Model Characteristics**   Model Size: The model includes 180 million parameters, occupying approximately 2.1 GB of disk space. Computational Complexity: The model has a computational cost of 1.63 GFLOPs (excluding visual feature extraction), making it feasible for implementation on robots without imposing excessive computational demands.

These metrics demonstrate that GR-DUET is highly practical for deployment in real-time systems. Its resource requirements align with the capabilities of robotics platforms such as TurtleBot2 and LoCoBot, which have been commonly used in previous works (Anderson et al., 2021; Xu et al., 2023).

Table 13: Variations in computational costs of GR-DUET across different episodes.

| Episode | 1 | 100 | 200 | 300 | 400 | 500 | 600 |
|---|---|---|---|---|---|---|---|
| Graph Coverage (%) | 4.1 | 68.1 | 81.1 | 89.6 | 94.1 | 97.4 | 98.5 |
| GPU memory (MB) | 823 | 2917 | 3609 | 3609 | 3609 | 3609 | 3609 |
| CPU memory (MB) | 5174 | 5252 | 5278 | 5284 | 5290 | 5291 | 5291 |
| Inference Time (ms) | 12 | 56 | 63 | 65 | 66 | 66 | 67 |

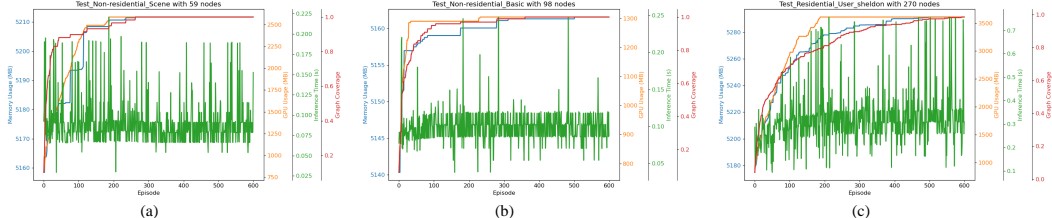

(a)                           (b)                           (c)

Figure 9: The change curve of GPU memory, CPU memory, inference time, and graph coverage in three environments with different sizes from GSA-R2R with episodes.

## A.5   MODEL SCALABILITY

In this section, we demonstrate the scalability of GR-DUET in larger and more complex environments, focusing on both computational efficiency and navigation performance.

**Computational Costs and Memory Usage**
Although the memory bank in GR-DUET updates continuously during navigation, it stabilizes once the agent has explored most of the environment. Updates after this point are minimal, involving only new instructions and actions, which require relatively little memory. Moreover, GR-DUET employs coarse-grained embeddings for nodes that are not neighbors of the current node, limiting GPU memory growth despite inputting the entire graph into the model. To illustrate this, we analyze key computational metrics across episodes for one of the largest environments in GSA-R2R, as shown in Tab. 13. As agents execute more instructions, we observe gradual increases in CPU memory usage, GPU memory usage, and inference time. However, when the graph coverage approaches 100%, indicating that the agent has explored most places, these metrics stabilize with minimal

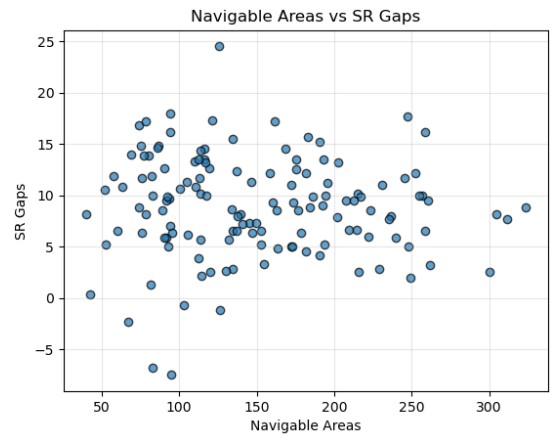

Figure 10: The SR gaps between DUET and our GR-DUET versus navigable areas in all 150 environments from GSA-R2R.

additional overhead. This demonstrates that GR-DUET is computationally scalable and does not sacrifice much memory efficiency for improved performance. Additionally, Fig. 9 presents the trends of these metrics across three environments of varying sizes from GSA-R2R, further supporting the scalability of GR-DUET. These results confirm that GR-DUET maintains its efficiency without compromising performance, even in large and complex navigation tasks.

**Navigation Performance**   In large environments, the memory bank grows, introducing more inputs to the model. However, GR-DUET's pre-training stage on the entire ground-truth graph ensures that it effectively handles these inputs. By focusing on surrounding viewpoints rather than the full graph, GR-DUET mitigates the impact of environment size on computational efficiency and performance. To validate this, we compare the performance gap (Success Rate, SR) between the original

Table 14: Mean SR gap between GR-DUET and original DUET by environment size.

| Viewpoint Number | 50-100 | 100-150 | 150-200 | 200-250 | 250-300 | >300 |
|---|---|---|---|---|---|---|
| Mean SR Gap (%) | 9.07 | 10.17 | 7.57 | 9.14 | 8.02 | 10.74 |

Table 15: Comparison of LLM-based methods and GR-DUET on GSA-R2R.

| Methods | Test-N-Basic | | Test-N-Scene | | Test-N-User | |
|---|---|---|---|---|---|---|
| | SR↑ | SPL↑ | SR↑ | SPL↑ | SR↑ | SPL↑ |
| MapGPT | 34.17 | 29.72 | 24.67 | 22.62 | 23.17 | 20.80 |
| NavCoT | 36.67 | 34.46 | 29.00 | 25.93 | 26.33 | 24.47 |
| NavGPT-2 | 63.50 | 47.26 | 56.67 | 43.34 | 47.00 | 36.86 |
| **GR-DUET (ours)** | **74.17** | **70.45** | **54.33** | **47.04** | **58.00** | **52.93** |

DUET and GR-DUET across all 150 environments in GSA-R2R with varying numbers of viewpoints in Tab. 14. The results show that GR-DUET consistently outperforms the original DUET, even in environments with a large number of viewpoints. This indicates that GR-DUET effectively learns to focus on relevant parts of the graph, ensuring scalability in large environments. Additionally, we include a scatter plot of SR gaps versus navigable areas in Fig. 10, which further confirms that GR-DUET consistently performs better in all large environments.

## A.6 PERFORMANCE OF LLM-BASED METHODS

Since today's LLMs have a strong capability of reasoning, we incorporate two additional LLM-based methods, MapGPT(Chen et al., 2024a) and NavCoT(Lin et al., 2024) to figure out whether they can adapt to different instruction styles without the need for an additional adaption process. Due to the scale of our GSA-R2R dataset, evaluating proprietary LLM-based methods on the entire dataset is computationally expensive. To address this, we sampled one environment for each type of instruction to conduct meaningful comparisons in Tab. 15. The results reveal that LLM-based methods perform poorly on the GSA-R2R task compared to GR-DUET. While LLMs can handle instructions with different styles, they struggle with the environmental adaptation required for GSA-VLN, particularly in processing visual information and interacting with persistent environments. We believe one promising direction for future research would be adapting LLM-based methods to specific environments using the memory bank provided by GSA-VLN.

We also test an intuitive idea of whether an LLM could translate styled instructions back into a basic style to facilitate the understanding of these styles for navigation models. This is evaluated on the Val-R-Scene split using three sets of instructions: 1. Basic: Instructions after Stage 2. 2. Scene: Instructions transformed from Basic after Stage 3.

Table 16: Performance comparison of instruction styles on the Val-R-Scene split.

| Instructions | Basic | Scene | Translated |
|---|---|---|---|
| SR | 46.37 | 42.30 | 44.83 |

3. Translated: Scene instructions translated back into Basic style by an LLM. The performance of these instruction types is summarized in Tab. 16. The results reveal that LLM-based translation improved performance over Scene instructions but does not fully close the gap with Basic instructions. This limitation arises from the open-vocabulary nature of LLMs, which introduces noise and leads to information loss, thereby reducing the effectiveness of the approach. It represents a promising direction for future work to solve the instruction style problem, like fine-tuning an LLM-based translator or adding the translated instructions into the training process of the navigation model.

## A.7 ADAPTATION EFFICIENCY

In this section, we analyze the adaptation efficiency of GR-DUET. GR-DUET builds a global graph for adaptation, which stabilizes once the agent has explored most parts of the environment. Based on the graph coverage data (as seen in Tab. 13), we find that 90% coverage is achieved with at most 400 instruction-trajectory pairs in large environments and as few as 100 pairs in small environments.

To measure adaptation speed, we treat the first $X$ instructions (the number required to reach 90% graph coverage) as the adaptation phase and divide them into groups of 50 instructions. Performance within each group is measured, and linear regression is applied to calculate the slope of performance improvement, serving as a proxy for adaptation speed. Results show that among the 150 scans, 94 achieved a positive slope, with a mean slope of 0.26, indicating rapid adaptation in most cases.

To understand slower or less effective adaptation, we analyzed adaptation speed across various environmental characteristics. First, Tab. 17 shows the mean adaptation slopes in environments with different numbers of floors. Adaptation becomes less effective as the number of floors increases, except for a few cases with four floors. This is intuitive, as distinct floor layouts and styles make prior memory from other floors less relevant to the current navigation. Conversely, Tab. 18 shows that adaptation efficiency improves as the number of rooms increases, particularly in buildings with more than 15 rooms. After viewing specific buildings, we find that environments with many rooms (e.g., hotels or student accommodations) often have repetitive layouts and identical rooms, allowing the agent to leverage memory from similar spaces effectively. These findings suggest that GR-DUET performs well in environments with repetitive structures but struggles in environments with dissimilar memory (e.g., multi-floor buildings with distinct layouts). We also calculate mean adaptation slopes for different scene and instruction types, as shown in Tab. 19. From the results, we can see that GR-DUET adapts faster in residential environments than in non-residential environments. This is likely due to the training environments being predominantly residential (from R2R), introducing a bias that favors residential scenarios. For different instructions, agents adapt fastest to Scene instructions and least effectively to User instructions. Scene instructions often include conversational fillers, which provide more distinct language patterns than the word variations in User instructions, making them easier to adapt to. These analyses highlight both the strengths and limitations of GR-DUET's adaptation mechanism. While it excels in environments with repetitive layouts, it struggles in multi-floor or irregular environments. Future improvements could focus on leveraging dissimilar memories (e.g., between floors) and reducing training biases to enhance adaptation in more diverse scenarios.

Table 17: The adaptation speeds of GR-DUET in environments with different number of floors.

| Floor Number | 1 | 2 | 3 | 4 |
|---|---|---|---|---|
| Mean Slopes | 0.24 | 0.19 | 0.05 | 0.40 |

Table 18: The adaptation speeds of GR-DUET in environments with different numbers of rooms.

| Room Number | 1-5 | 5-10 | 10-15 | 15-20 | 20-25 | 20-25 | >25 |
|---|---|---|---|---|---|---|---|
| Mean Slopes | 0.48 | 0.04 | 0.06 | 0.20 | 0.28 | 0.36 | 0.70 |

Table 19: The adaptation speeds of GR-DUET in different types of scenes and instructions.

| Type | Residential | Non-residential | Basic Inst. | Scene Inst. | User Inst. |
|---|---|---|---|---|---|
| Mean Slopes | 0.34 | 0.18 | 0.21 | 0.55 | 0.19 |

## A.8 Human Study Details

In this section, we provide more details about the human study conducted in Tab. 2 to prove its reliability. Our study included 15 participants, comprising university students and staff aged between 20 and 35 years old. The participants represented a diverse range of genders and backgrounds, ensuring a variety of perspectives while maintaining a degree of homogeneity necessary for unbiased evaluations. Two tasks were given to the participants and both of the tasks were straightforward and did not require specialized knowledge. In task 1, the participants need to determine whether an instruction aligns with the corresponding trajectory. Task 2 requires them to assess whether the instruction exhibits a distinct speaking style. Given the simplicity of these tasks, we are confident that our participants were competent to perform the evaluation accurately and reliably. We provide the example figure of the user interface used in the human study in Fig. 11.

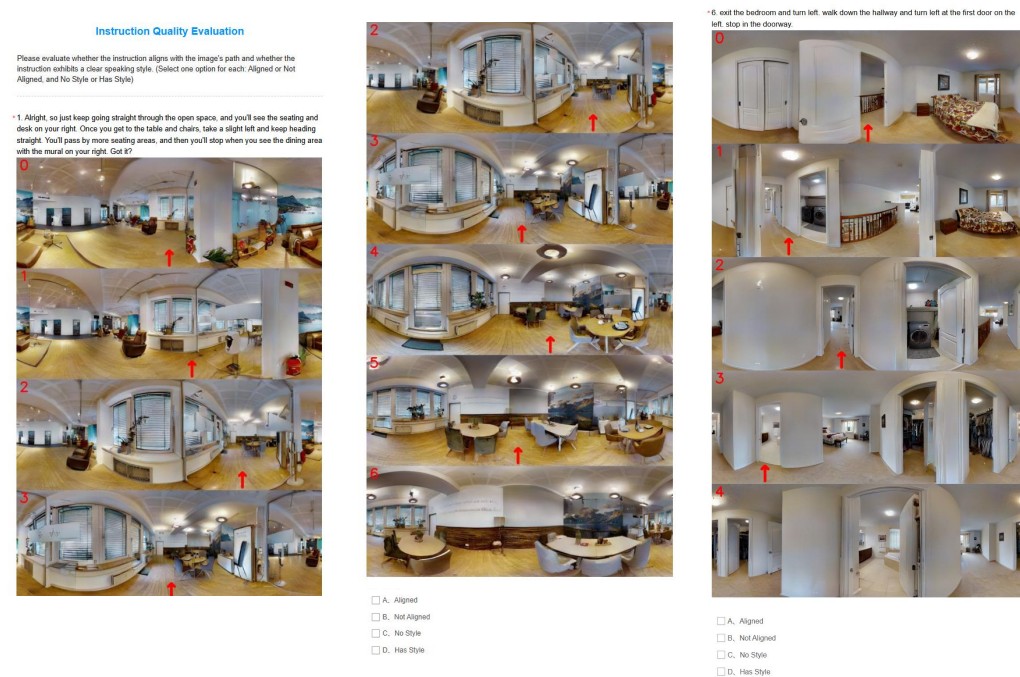

Figure 11: Examples of the problems for the participants in the human study.

## A.9 INSTRUCTION DIVERSITY ANALYSIS

Although we only generate User instructions with only five characters, our approach can be extended to generate instructions for thousands of characters. Specifically, by utilizing the SummScreen dataset (Chen et al., 2022a), our method can scale up to approximately 3,000 unique characters, which we believe provides sufficient diversity for research purposes. Since our work is the first to propose the problem of instruction style adaptation, the primary aim is to establish the existence of instruction adaptation as a problem and provide a meaningful benchmark to test the adaptation capabilities of current methods.

To achieve these goals, we find that using five characters appears to be sufficient, as evidenced by the results in Tab. 5. To further justify this, we calculate the word overlap rate between the instructions of the five selected characters and the remaining 173 candidate characters. This calculation is based on five example instructions. The results in Tab. 20 demonstrate that our selected characters already cover a broad range of language patterns. Adding more characters would slightly increase the dataset's scope but is unnecessary for addressing the core problem or establishing a robust benchmark. Notably, the character with the least overlap rate still shares a 65% overlap with our selected characters, highlighting the diminishing returns of including additional characters. For comparison, in the widely used R2R dataset (Anderson et al., 2018b), the evaluation split includes only 11 scans, far fewer than would exist in real-world scenarios. Yet, it is broadly accepted as a standard benchmark for evaluating navigation capabilities. Similarly, our choice of five characters maintains practicality while ensuring meaningful evaluation. Moreover, our dataset includes Scene instructions, which encompass styles influenced by professional language and roles, further broadening the diversity of speaking styles. This additional dimension ensures that the five selected User instruction styles are adequate for evaluating adaptation methods.

Table 20: Word overlap rate between our characters and the remaining 173 characters.

| Instruction | 1 | 2 | 3 | 4 | 5 | Average |
|---|---|---|---|---|---|---|
| Mean Overlap | 0.90 | 0.93 | 0.81 | 0.90 | 0.82 | 0.87 |

---

**Prompt Template for Generating Basic Instructions**

**System Instruction:**

You are an expert in annotating instructions for navigation paths.

The path is presented in two formats:

1. A sequence of 360-degree panoramic observations from the start to the goal location, each with a red arrow indicating the direction of movement and a red number on the top left corner indicating the step number, ending with a panorama at the goal location.

2. A map image with green blocks representing discrete viewpoints and red arrows indicating movement directions. Gray areas on the map indicate possible movement regions.

Given the instruction: {initial instruction}

Your tasks are:

1. Identify any errors or violations in the instruction based on the following criteria:
   - The instruction must be concise, natural, and complete without being split into steps.
   - The instruction must specify a unique, unmistakable goal location.
   - The instruction must avoid any ambiguity that might mislead the robot.
   - The instruction must match exactly the map, especially the directions of movement.

2. Improve the instruction based on your analysis so that a smart robot can find the goal location after starting from the same start location using only the improved instruction.

You will be penalized if the new instruction does not satisfy the criteria.

Provide the analysis and improved instruction directly without additional commentary in the format: Analysis: xxx. Improved Instruction: xxx.

**User Prompt:**

{panoramas along the path}; {top-down map image of the path}

Figure 12: Prompt template for classifying buildings from HM3D.

As for the specific characters, we selected them based on their highly distinct speaking styles, ensuring diversity in age, gender, and language patterns. To measure the diversity of speaking styles, we generated rephrased instructions for each character using the same five Basic instructions and calculated the word overlap rate. Lower overlap rates indicate more distinct language patterns. These results, along with the characters' rankings among all characters in SummScreen, are presented in Tab. 21. The overlap rates and rankings confirm that the chosen characters exhibit distinct and diverse language patterns, making them ideal representatives for User instructions. This diversity ensures that our dataset effectively challenges VLN models to adapt to different speaking styles.

Table 21: Word overlap rate and ranking of four selected characters in GSA-R2R among all characters in SummScreen.

| Character | Keith | Moira | Rachel | Sheldon |
|-----------|-------|-------|--------|---------|
| Overlap Rate | 0.44 | 0.28 | 0.42 | 0.30 |
| Ranking | 8th | 1st | 6th | 2nd |

### A.10 PROMPT TEMPLATES

In this section, we provide the prompt templates used in our instruction orchestration pipeline:

- Fig. 12: Prompt template for classifying buildings from HM3D;

- Fig. 13: Prompt template for refining speaker-generated instructions into *Basic* instructions (stage 2 of the instruction orchestration pipeline);

- Fig. 14: Prompt template for rephrasing *Basic* instructions into *Scene* instructions;

- Fig. 15: Prompt template for rephrasing *Basic* instructions into *User* instructions (excluding Child);

- Fig. 16: Prompt template for rephrasing *Basic* instructions into *User* instructions with the Child Character;

We directly use the prompt templates for generating role files and dialogue history from RoleLLM (Wang et al., 2023a).

---

**Prompt Template for Classifying Buildings**

**System Instruction:**

You are a helpful assistant in determining the building type according to the provided images of the building from the following candidate types: {187 building types}.
All images refer to the same building.
Only give me the building type without additional information.

**User Prompt:**

{overview image for the building}; {top-down views for each floor}; {representative panoramas}

---

Figure 13: Prompt template for refining speaker-generated instructions into *Basic* instructions.

---

**Prompt Template for Generating Scene Instructions**

**System Instruction:**

I will provide a navigation instruction and the corresponding building type. Your task is to transform the instruction to match the speaking style appropriate for the specified building type and its typical users or visitors. Follow these steps:
    1. Identify typical users or visitors of the building type (e.g., hotel: hotel owner, waiter, guest).
    2. Randomly select one of these users or visitors as the speaker.
    3. Modify the instruction to match the selected speaker's conversational style, incorporating minor elaborations and conversational fillers (e.g., 'you know', 'um', 'let's see', etc.) to enhance context and make it **as distinct as possible from the original.**

    Important Notes:
- Do not alter any navigation details or specific information; the core directions must remain unchanged.
- Avoid explicitly mentioning the audience in the instruction (e.g., for a teacher, do not say 'Hi, students').

    Return Format:
1. Potential Speakers: List of potential speakers
2. Chosen Speaker: The selected speaker's role
3. New Instruction: The modified instruction
4. Reason for Modification: Explanation of how the changes reflect the chosen speaker's style

**User Prompt:**

Instruction: {basic instruction}
Building Type: {building type}

---

Figure 14: Prompt template for rephrasing *Basic* instructions into *Scene* instructions.

## A.11 DISCUSSION

**Limitations.** Despite using state-of-the-art GPT4o to refine speaker-generated instructions, some errors remain in the output instructions due to the difficulty VLMs face in understanding spatial relationships between panoramas. Another limitation is that our GR-DUET method focuses on scene adaptation mainly from a visual perspective by utilizing previous observations, but lacks a language-specific design to capture the consistent speaking style in a persistent environment. Although we experimented with several methods to address this, they were unsuccessful. Lastly, while this paper addresses the general scene adaptation problem with step-wise instructions, real-world navigation often involves other instruction types, such as object-oriented or dialog-based instructions, which we have not covered here.

**Future Work.** In the future, we plan to address the mentioned limitations to make our task more general, incorporating datasets with diverse instruction types, such as those in REVERIE (Qi et al., 2020) and CVDN (Thomason et al., 2020). We will also explore ways to enhance the panorama understanding capabilities of VLMs to improve their comprehension of path observations to generate instructions that are comparable to human-annotated ones. In terms of methodology, we will incorporate more unsupervised learning approaches, with a key focus on linking their optimization objectives to navigation performance, such as combining proxy tasks with back-translation methods.

---

**Prompt Template for Generating User Instructions Excluding Child**

**System Instruction:**

You are {character} from the TV series {show}, {role profile} {catchphrase}. Now please answer some questions to accurately show your personality traits! Your speaking style should fully imitate the personality role assigned to you! Please do not expose that you are an artificial intelligence model or a language model, you must always remember that you are only assigned one personality role. Don't be verbose or too formal or polite when speaking.

**User Prompt:**

{line_1 from the character}

**Assistant Prompt:**

{lines from other characters}

**User Prompt:**

{line_2 from the character}

**Assistant Prompt:**

{lines from other characters}

….

**User Prompt:**

Hey {character}, I need to give some directions, but I want them to sound like they are coming from you, with your unique style, and **as different as possible from the original**. Here is the instruction:{basic instruction}

---

Figure 15: Prompt template for rephrasing *Basic* instructions into *User* instructions (excluding Child).

---

**Prompt Template for Generating User Instructions of Child Character**

**System Instruction:**

I will provide a navigation instruction. Your task is to rewrite the instruction as if it is being spoken by a child. The revised instruction must capture the tone, language, and mannerisms typical of a child's speech.

Guidelines:
- Do not alter any navigation details or specific information; the core directions must remain unchanged.
- Use simple, child-friendly language with shorter sentences, a playful tone, and add child-like expressions, exclamations, and avoid jargon or complex words.
- Make the new instruction **distinctly different from the original.**

Output Format:
1. New Instruction: The version of the instruction as spoken by a child.
2. Reason for Modification: A brief explanation of how the changes reflect the style of a child.

**User Prompt:**

Instruction:{basic instruction}

---

Figure 16: Prompt template for rephrasing *Basic* instructions into *User* instructions with the Child Character.

