# OpenReview forum: "General Scene Adaptation for Vision-and-Language Navigation"
_ICLR.cc/2025/Conference — ICLR 2025 Poster_

### Official Review · Reviewer_J7gj · 2024-10-28

**Soundness:** 3
**Presentation:** 3
**Contribution:** 2
**Rating:** 6
**Confidence:** 4

**Summary:**

The paper presents a novel task, GSA-VLN (General Scene Adaptation for Vision-and-Language Navigation), which trains agents to follow navigation instructions within a specific scene while adapting to it for enhanced performance over time. A new dataset, derived from the HM3D dataset, is introduced to support this task. Additionally, the authors propose a new method that serves as a baseline and achieves state-of-the-art performance in this domain.

**Strengths:**

- The paper is generally well motivated and the proposed tasks makes sense. For a pre-trained VLN agent, it is important to leverage the information and instructions in the new environment to further enhance it's knowledge and adapt to the new environment and uses.

- A new dataset GSA-R2R based on the HM3D dataset is introduced with new instruction data collected to support the VLN task. The dataset can potentially be useful for the community.

-  Extensive evaluation of current VLN methods on the new dataset and different adaption methods are benchmarked. The proposed GR-DUET method demonstrates competitive performance compared to prior work.

**Weaknesses:**

- The concept of adapting a model to a test scene is not entirely new, as numerous prior methods have explored unsupervised exploration or adaptation within embodied environments.

- The Related Work section could be more comprehensive. For instance, some discussions are postponed to Section 4, but it's crucial to review prior work that also employs adaptation methods in VLN, particularly those utilizing memory-based approaches, and also highlight the main differences and contributions of the proposed method.

- Additionally, beyond the VLN literature, how does the proposed method relate to Lifelong Learning and Test-time Adaptation?

- Table 3 presents the navigation performance of various VLN models on the new dataset. Is the performance of these methods consistent with results on other benchmark datasets?

**Questions:**

Please see above.

---

> ### Author Response · Authors · 2024-11-22
> **Response to Reviewer J7gj - Part 1**
>
> We greatly appreciate Reviewer J7gj for the thorough review of our paper and for providing highly constructive comments. Our responses are detailed below.
>
> ---
>
> > ### Weakness 1. Adaptation Novelty
>
> We acknowledge that the concept of adaptation has been explored in embodied environments, and we have discussed related works in the second and third paragraphs of the *Related Work* section. However, scene adaptation remains an underexplored area within VLN, and our work seeks to address this gap.
>
> ### **1.1 How Our Work Differs from Related Efforts**
>
> The only directly related work in VLN is Iterative VLN (IVLN), which we have discussed in Lines 51–53 and 73–78 of the manuscript. Our approach differs from IVLN in two key ways:
>
> 1. **Enabling Both Memory-Based and Adaptation-Based Methods**: We include a significantly larger number of instructions per scene, allowing agents to leverage both memory-based and adaptation-based methods for scene-specific tasks.
> 2. **Introduction of OOD Data**: While IVLN only utilizes the original R2R dataset, which is limited in scene and instruction diversity, our work incorporates both ID and OOD data which can better evaluate agent adaptability and align more closely with real-world scenarios.
>
> ### **1.2 Novel Contributions of Our Work**
>
> Although the concept of adaptation itself is not entirely new, our work introduces several novel contributions that address previously overlooked aspects of VLN research:
>
> 1. General Scene Adaptation Task: We are the first to propose the general scene adaptation task for VLN, addressing both ID and OOD scenes and instructions.
>
> 2. Largest Indoor VLN Dataset: Our GSA-R2R dataset is the largest indoor VLN dataset to date, comprising 150 scenes and 90K instructions (Table 1 of the paper).
>
> 3. Diverse Environments and Instructions: We expand the range of environments by incorporating 20 scene types, including both residential and non-residential buildings. We also introduce diverse speaking styles, opening a new research direction in VLN that investigates the impact of instruction diversity on navigation performance.
>
> We believe these contributions are critical for advancing VLN research and provide a foundation for future work in this area.
>
> ---
>
> >### Weakness 2. Related Work Scope
>
> Thank you for your valuable suggestion. We appreciate your feedback regarding the structure and scope of the *Related Work* section.
>
> ### **2.1 Reason for Original Structure**
>
> Since our primary focus is on proposing the GSA-VLN task and the GSA-R2R dataset, we initially deferred detailed descriptions of related adaptation methods to Section 4 alongside the introduction of our GR-DUET method. Similarly, key differences and contributions were outlined in Lines 363–367 within the method section to maintain consistency with the experimental context.
>
> ### **2.2 Revisions Made**
>
> However, we agree that discussing related methods and their differences from our approach should be included in the *Related Work* section to enhance clarity and accessibility. Therefore, we have reorganized the structure of the paper with two modifications:
>
> 1. We have integrated a detailed discussion of adaptation methods and memory-based approaches in VLN, including how they address environment-specific challenges and their limitations in our task.
> 2. We explicitly highlighted the key differences between our proposed method and existing memory-based works, including the global topological map for enhancing spatial understanding of history and the specialized training strategy for addressing the input distribution shift problem.
>
> We hope these changes address your concern and enhance the paper’s accessibility. Thank you again for your constructive feedback.
>
> ---
>
> >### Weakness 3. Lifelong Learning and TTA
>
> Thank you for your insightful comment. Our work indeed shares certain similarities with Lifelong Learning and Test-Time Adaptation (TTA), as all these settings involve continuous learning. However, there are distinct differences that set our approach apart.
>
> ### **3.1 Comparison with Lifelong Learning**
>
> Lifelong Learning focuses on acquiring new skills or tasks over time while preserving the ability to perform previously learned tasks [1]. In contrast, our setting involves an agent adapting to a single scene over its lifetime. Rather than acquiring and retaining multiple skills or tasks, our work emphasizes **repeated mastery of the same navigation skill** within a scene-specific context. This makes our approach distinct from the broader objective of lifelong learning.

---

> ### Author Response · Authors · 2024-11-22
> **Response to Reviewer J7gj - Part 2**
>
> ### **3.2 Comparison with TTA**
>
> Our work is more closely related to TTA, as we both aim to adapt the agent during inference without additional supervision. Several TTA methods, such as TENT and SAR, are included as baselines in our experiments to evaluate their applicability to the GSA-VLN task. While TTA provides a viable solution to the scene adaptation problem, our approach goes beyond TTA by incorporating memory-based methods, where model parameters remain fixed during adaptation but the input varies based on the dynamically updating memory bank. This integration of memory-based methods demonstrates the broader applicability and versatility of our setting compared to TTA.
>
> To further clarify this, we have added Section A.9 in the revised manuscript to discuss the relationship between our work and these two related areas. Thank you for allowing us to improve our manuscript.
>
> &nbsp;
>
> [1] Liu, Bing. "Lifelong machine learning: a paradigm for continuous learning." *Frontiers of Computer Science* 11.3 (2017): 359-361.
>
> ---
>
> > ### Weakness 4. Benchmark Consistency
>
> Thank you for your valuable suggestion. We appreciate the opportunity to clarify the consistency of model performance across datasets and the rationale behind the presentation in Table 3.
>
> ### **4.1 Purpose of Table 3**
>
> Table 3 is primarily designed to illustrate the unique characteristics of our GSA-R2R dataset, including:
>
> 1. **Scene Diversity**: The comparison between residential and non-residential scenes.
> 2. **Instruction Types**: The impact of different instruction styles (e.g., Basic, Scene, and User).
>
> Given that other VLN benchmark datasets, such as R2R, focus only on residential scenes with basic instructions, their evaluation conditions align with our **Test-R-Basic** split in GSA-R2R. For this reason, we initially chose not to include performance on other datasets in Table 3.
>
> ### **4.2 Revisions to Enhance Completeness**
>
> We agree that adding the R2R performance of each baseline can enhance the completeness of the comparison. Since we have already referenced these results in Line 428, we have now added a column to Table 3 in the revised manuscript to present the R2R performance alongside GSA-R2R results for comparison.
>
> ### **4.3 Performance Across Datasets**
>
> Comparing the performance of R2R and our GSA-R2R, we provide the following observations:
>
> - Ranking Consistency: The ranking of baseline performance is consistent across datasets, with ScaleVLN achieving the highest scores and HAMT the lowest.
>
> - Large Performance Gap: Specific performance numbers are significantly lower on GSA-R2R compared to R2R. This performance drop reflects the increased complexity of GSA-R2R, which includes more diverse scenes, complex paths, and challenging instructions. These results highlight the additional challenges posed by GSA-R2R and its value in evaluating methods under more realistic and diverse conditions.
>
> We hope these revisions and clarifications address your concerns and provide a more complete understanding of the benchmarks used in our study. Thank you again for your constructive feedback.

---

> > ### Comment · Reviewer_J7gj · 2024-11-25
> >
> > Thank the authors for the detailed response, and most of my questions are addressed. I have also read other reviews and generally agree with my fellow reviewers. Therefore, I am keeping my original score.

---

> > > ### Author Response · Authors · 2024-11-26
> > > **Response to Reviewer J7gj**
> > >
> > > Thank you for taking the time to review our responses and for your thoughtful consideration of our work. We are glad to hear that most of your questions have been addressed and that our clarifications were helpful. Your constructive feedback has been instrumental in refining our work, and we value your thoughtful insights throughout the review process.
> > >
> > > Thank you once again for your time and effort in reviewing our submission!

---

### Official Review · Reviewer_J6Xq · 2024-10-30

**Soundness:** 2
**Presentation:** 2
**Contribution:** 2
**Rating:** 6
**Confidence:** 3

**Summary:**

This paper proposes a new task, named GSA-VLN, which require agents to execute navigation instructions within a specific scene and simultaneously adapt to it for improved performance over time. This paper also proposes a new datast, GSA-R2R, which significantly expands the diversity and quantity of environments and instructions for the Room-to-Room (R2R) dataset to evaluate agent adaptability in both ID and OOD contexts. The biggest difference between the proposed task and dataset and previous work is the diversity of instructions, i.e., different individual features and linguistic conventions are taken into account.

**Strengths:**

1. The proposed GSA-VLN task and  GSA-R2R dataset which considers real-world robot adaptation in persistent environments, is an interesting research direction.
2. Overall, the writing is fluent and the figures convey the character of the task well.
3. The proposed GR-DUET method outperforms the baselines, demonstrating its effectiveness in helping agents adapt to specific environments.

**Weaknesses:**

1. Some baselines are missing. I suggest to add some baseline methods based on LLM, especially the Zero-shot VLN methods. For example, InstructNav[1], NavCot[2]. The reason for this suggestion is that LLM's reasoning is now so powerful that it may be able to adapt well for different personal characteristics and language styles without the need for an additional adaption process, i.e., in zero-shot manner.  Also, these different styles are essentially generated by LLM, so I'm concerned that these understanding these different styles is a very easy and undifferentiated thing for LLM to do.
2. In this paper, the authors generated instructions for only five different character styles. However life can be much richer in terms of characters. The paper's contribution would have been greatly enhanced if the authors could propose a method for generating instructions for different character styles in a nearly infinite number of ways.
3. The authors propose GR-DUET, but there are few details about it. For a reader who does not know DUET well, it may cause some difficulties in reading, and I suggest the authors to add some descriptions and details in the appendix.



[1] InstructNav: InstructNav: Zero-shot System for Generic Instruction Navigation in Unexplored Environment.

[2] NavCoT: Boosting LLM-Based Vision-and-Language Navigation via Learning Disentangled Reasoning.

**Questions:**

In section 3.3.4, the authors invite 15 participants to evaluate the instructions. Are the backgrounds (e.g., ages, etc.) of these 15 participants sufficiently homogeneous to demonstrate the refinement of the assessment? Also, I recommend disclosing the questionnaire they used for the test.

---

> ### Author Response · Authors · 2024-11-22
> **Response to Reviewer J6Xq - Part 1**
>
> We thank Reviewer J6Xq for taking the time to review our paper and for the constructive feedback provided. Please refer to the following responses to address your comments.
>
> ---
>
> > ### Weakness 1. LLM-based Baselines
>
> Thank you for the suggestion to include LLM-based methods as baselines.
>
> ### **1.1 Existing LLM-based Baselines in Our Experiments**
>
> Our experiments already include **NavGPT2**, a recent LLM-based method from ECCV 2024. As shown in Table 3 of the paper, NavGPT2's performance reveals that LLM-based methods, when applied in a zero-shot manner without adaptation techniques, do not exhibit significant advantages for the scene adaptation problem.
>
> ### **1.2 Adding More LLM-based Baselines**
>
> We are happy to include additional LLM-based methods in our evaluation. While InstructNav is a relevant VLN method, it operates in **continuous** VLN settings with low-level actions (e.g., "move forward" or "turn around"). Our GSA-VLN task, on the other hand, focuses on **discrete** environments. This fundamental difference makes a direct comparison challenging.
> Therefore, we incorporate another recent LLM-based method, MapGPT [1], which has demonstrated strong zero-shot capabilities, along with NavCoT, for evaluation.
> Due to the scale of our GSA-R2R dataset, evaluating proprietary LLM-based methods on the entire dataset is computationally expensive. To address this, we sampled one environment for each type of instruction to conduct meaningful comparisons. The results are summarized in **Table 1** below:
>
> **Table 1: Comparison of LLM-based methods and GR-DUET on GSA-R2R.**
>
> | **Methods**        | **Test-N-Basic (SR ↑)** | **Test-N-Basic (SPL ↑)** | **Test-N-Scene (SR ↑)** | **Test-N-Scene (SPL ↑)** | **Test-N-User (SR ↑)** | **Test-N-User (SPL ↑)** |
> | ------------------ | ----------------------- | ------------------------ | ----------------------- | ------------------------ | ---------------------- | ----------------------- |
> | MapGPT             | 34.17                   | 29.72                    | 24.67                   | 22.62                    | 23.17                  | 20.80                   |
> | NavCoT             | 36.67                   | 34.46                    | 29.00                   | 25.93                    | 26.33                  | 24.47                   |
> | NavGPT-2           | 63.50                   | 47.26                    | 56.67                   | 43.34                    | 47.00                  | 36.86                   |
> | **GR-DUET (ours)** | **74.17**               | **70.45**                | **54.33**               | **47.04**                | **58.00**              | **52.93**               |
>
> The results reveal that LLM-based methods perform poorly on the GSA-R2R task compared to GR-DUET. While LLMs can handle instructions with different styles, they struggle with the environmental adaptation required for GSA-VLN, particularly in processing visual information and interacting with persistent environments.
>
> We believe one promising direction for future research would be adapting LLM-based methods to specific environments using the memory bank provided by GSA-VLN. Techniques such as Retrieval-Augmented Generation (RAG) or Parameter-Efficient Fine-Tuning (PEFT) may enhance the ability of LLM-based methods to handle scene-specific challenges in tasks like GSA-VLN.
>
> Thank you again for the suggestion, and we have included these results and discussions in Section A.6 of the revised manuscript.
>
> &nbsp;
>
> [1] Chen, Jiaqi, et al. "Mapgpt: Map-guided prompting with adaptive path planning for vision-and-language navigation." ACL. 2024.
>
> ---
>
> > ### Weakness 2. More Diverse Styles
>
> Thank you for your suggestion regarding the diversity of character styles. We would like to address a potential misunderstanding and clarify the scalability of our approach.
>
> ### **2.1 Scalability of Our Method**
>
> **We do propose a method for generating instructions for different character styles, which can be highly scalable**.
> As detailed in Lines 885–894 of the paper, our approach can be extended to generate instructions for thousands of characters. Specifically, by utilizing the SummScreen dataset, our method can scale up to approximately 3,000 unique characters, which we believe provides sufficient diversity for research purposes.

---

> > ### Comment · Reviewer_J6Xq · 2024-11-24
> >
> > Thanks for the authors' reply. I thank the authors for providing information about DUET and the Human Study Details.
> > However, I still have some concerns.
> >
> > 1. Continuous or discrete doesn't seem to affect the comparisons entirely, especially when the real world environment is all continuous, and I think continuous might be more important. I still suggest that the idea of InstructNav can be modified for some comparisons (it doesn't have to be an identical reproduction).
> >
> > 2. The very puzzling point for me is that since the dataset is generated by LLM, why doesn't LLM perform well on it? That's why I highly recommend comparing InstrucNav, which requires no training at all and **consists purely of LLMs**.
> >
> > 3. GR-DUET doesn't  seem to have done something to design for the different styles in GSA dataset, which makes me think there is some separation between the method and the GSA  dataset. Moreover, is memory a solution for dealing with different styles? If GR-DUET has to handle a very large number of different styles at the same time, does its performance degrade, since the size of memory bank is limited? For example, the VLN agent is placed on the first floor to receive different customers.
> >
> > 4.  "Why We Selected Five Characters". I'm not really convinced by the author's reasoning. This seems to be a subjective judgment on the part of the authors, not a result based on an objective experiment results.

---

> > > ### Author Response · Authors · 2024-11-24
> > > **Response to Reviewer J6Xq - Part 1**
> > >
> > > Thanks for your valuable feedback. We answer your concerns as follows:
> > >
> > > ---
> > >
> > > > ### Question 1: InstructNav Reproduction
> > >
> > > We would like to address the exclusion of InstructNav from our comparisons from the following perspectives:
> > >
> > > ### **1.1 Discrete vs. Continuous Navigation**
> > > Our work specifically targets a novel task and dataset in the discrete setting, which fundamentally differs from continuous navigation. Key distinctions include:
> > >
> > > - Perception: Panoramic views (discrete) vs. single-view perception (continuous).
> > > - Actions: High-level (discrete) vs. low-level actions (continuous).
> > > - Auxiliary Inputs: Discrete navigation does not rely on depth or semantic segmentations, which are essential in continuous setups.
> > >
> > > These distinctions make comparisons between discrete and continuous settings neither straightforward nor entirely relevant, as the underlying requirements, task formulations, and challenges differ greatly. While the continuous environment setting is indeed important, the focus of our work is not on completing navigation in continuous environments. Instead, our primary goal is to explore how agents can progressively adapt to their environments. Therefore, our current research advances the discrete VLN setting.
> > >
> > > ### **1.2 Incomplete Code Availability**
> > > InstructNav has only open-sourced its implementation for ObjectNav, with no code available for VLN-CE (as noted in this [issue](https://github.com/LYX0501/InstructNav/issues/6#issuecomment-2421016097)). Reproducing or adapting InstructNav for our discrete setting would require significant effort, including addressing the following challenges:
> > >
> > > 1. Panoramic Views: Designing prompts and pipelines to handle panoramic views instead of single-view inputs.
> > > 2. Auxiliary Inputs: Modifying the approach to function without depth and semantic information, which are not used by other baselines in our work.
> > > 3. Action Spaces: Adapting from low-level continuous actions to high-level discrete actions.
> > >
> > > If we were to forcibly implement InstructNav, such changes would require extensive redesign and could lead to unpredictable performance variations. This would make comparisons with other methods unreliable, potentially rendering the comparison of InstructNav meaningless in this context.
> > >
> > > ### **1.3 Existing LLM-Based Baselines**
> > > Furthermore, if the reviewer is specifically interested in comparisons with methods that do not require training and are composed of LLMs, our work already includes three representative LLM-based baselines:
> > > - NavGPT2 \& NavCoT: A fine-tuned LLM for navigation tasks.
> > > - MapGPT: A zero-shot method that relies purely on LLMs without additional training.
> > > All these methods are specifically designed for the discrete setting, providing more relevant and compelling comparisons for our work.
> > >
> > > While we acknowledge that InstructNav is an excellent zero-shot navigation system, its core ideas, such as dynamic navigation chains and multi-source value maps, do not directly align with the objectives of scene-specific adaptation tasks. We have included InstructNav in the Related Work section to provide a more comprehensive discussion.
> > >
> > >
> > > ### **1.4 Focus on Adaptation Over Zero-Shot Performance**
> > > Our work focuses on scene adaptation, not zero-shot navigation. Even if LLM-based methods achieve strong performance in our task (which is not true), they do not incorporate adaptation techniques for specific environments, which is central to our study. Without adaptation mechanisms, LLM-based methods resemble baseline VLN models like DUET in their approach. This focus on adaptation also explains why Table 3 in the paper is dedicated to illustrating dataset characteristics rather than comparing baseline VLN models. Our primary interest lies in the results presented in Tables 4–6, which evaluate the SR improvements achieved through adaptation-based and memory-based methods over the original DUET baseline, rather than the absolute baseline performance of different VLN models. By highlighting these improvements, we aim to emphasize the impact of adaptation techniques in our proposed task.
> > >
> > > While we recognize the strengths of InstructNav in the context of zero-shot navigation, it falls outside the scope of our work. Our goal is to establish a foundation for scene adaptation in discrete navigation tasks, encouraging the research community to build upon this framework. As demonstrated, current LLM-based methods struggle with our proposed task, highlighting opportunities for future research. Adapting InstructNav or similar LLM-based methods to our setting could be a valuable direction for addressing scene adaptation but is not the focus of our study.

---

> > > ### Author Response · Authors · 2024-11-24
> > > **Response to Reviewer J6Xq - Part 3**
> > >
> > > > ### Question3: GR-DUET and Memory Mechanism
> > >
> > > ### **3.1 Method and the Dataset**
> > >
> > > As mentioned in the common response to all reviewers, the core contribution of our work lies in introducing the novel GSA-VLN task and the corresponding GSA-R2R dataset. This dataset emphasizes the challenges faced by VLN agents in adapting to persistent environments that include both ID and OOD scenarios and diverse instruction styles. Our primary focus is on framing the problem, proposing this new task setting, generating a high-quality, diverse dataset, and evaluating existing methods under these conditions.
> > >
> > > While we propose the GR-DUET method as part of our work, it is primarily intended as a baseline—a proof of concept that provides an initial and feasible approach to tackling this task. This baseline establishes a foundation for further exploration and refinement by the research community. By doing so, our paper aims to catalyze the development of more sophisticated methods for addressing this realistic and practical setting.
> > >
> > > To clarify, there is no separation between the methods and the dataset. Our work evaluates existing methods, such as Masked Language Modeling (MLM) and Back-Translation (BT), under the GSA-VLN setting, demonstrating how these methods adapt to diverse instruction styles and persistent environments. This ensures our paper is complete as a dataset paper while also introducing a baseline method for further exploration. Instruction adaptation remains an open and promising research direction, enabled by the benchmarks and resources we provide.
> > >
> > > ### **3.2 Memory as a Solution**
> > >
> > > We would like to clarify the role of the memory bank in GR-DUET. The memory bank is a key component designed to store historical information for adaptation purposes. It is not fixed in size and can be updated dynamically. To address concerns about scalability, we provide evidence in Table 2 that the memory mechanism remains efficient as the number of episodes increases. The data shows that computational costs, including GPU and CPU memory usage and inference time, do not become a bottleneck even as the memory bank grows.
> > >
> > > **Table 2: Variations in computational costs of GR-DUET across different episodes.**
> > >
> > > | Episode             |      1      |     100     |     200     |     300     |     400     |     500     |     600     |
> > > | ------------------- | ----------- | ----------- | ----------- | ----------- | ----------- | ----------- | ----------- |
> > > | Graph Coverage (%)  |     4.1     |    68.1     |    81.1     |    89.6     |    94.1     |    97.4     |    98.5     |
> > > | GPU memory (MB)     |     823     |    2917     |    3609     |    3609     |    3609     |    3609     |    3609     |
> > > | CPU memory (MB)     |    5174     |    5252     |    5278     |    5284     |    5290     |    5291     |    5291     |
> > > | Inference Time (ms) |     12      |     56      |     63      |     65      |     66      |     66      |     67      |
> > >
> > > Regarding the question of whether the memory bank is a definitive solution for handling different instruction styles, we acknowledge that this remains an open research question. Our experiments demonstrate that utilizing the memory bank with TTA, back-translation, and MLM methods leads to performance gains, which we find encouraging. We hope our work inspires future research to develop better mechanisms for leveraging the memory bank and achieving superior performance.
> > >
> > > ### **3.3 Handling a Large Number of Styles Simultaneously**
> > >
> > > The scenario of handling a "very large number of different styles at the same time" is beyond the scope of this work. Our task focuses on scene adaptation within environments where instruction styles and users are consistent.
> > >
> > > For scenarios like  "a VLN agent placed on the first floor to receive different customers," they represent a different problem, which falls under the scope of traditional VLN rather than GSA-VLN. In this case, customer changes frequently, and adapting to such changes is outside the intent of our proposed task, which is rooted in persistent environmental adaptation. We believe addressing such scenarios requires different approaches, distinct from the objectives and methods outlined in our work.

---

> > > ### Author Response · Authors · 2024-11-24
> > > **Response to Reviewer J6Xq - Part 4**
> > >
> > > > ### Question 4: The Selection of Characters
> > >
> > > ### **4.1 Addressing the Reviewer’s Original Comment**
> > > The original review mentioned that "the paper's contribution would have been greatly enhanced if the authors could propose a method for generating instructions for different character styles in a nearly infinite number of ways." In response, we demonstrated that our method is capable of generating instructions for thousands of character styles, leveraging datasets such as SummScreen. This scalability highlights the robustness and flexibility of our approach, which addresses the review's concern about the generalizability of our method.
> > >
> > > ### **4.2 Why Five Characters?**
> > > Since our work is the first to propose the problem of instruction style adaptation, the primary aim is to:
> > >
> > > - Establish the existence of instruction adaptation as a problem: Demonstrating that different speaking styles significantly affect the performance of VLN models.
> > > - Provide a meaningful benchmark: Offering data to test the adaptation capabilities of current methods.
> > >
> > > To achieve these goals, we find that using five characters appears to be sufficient, as evidenced by the results in Table 5 of the paper. To further justify this, we calculate the word overlap rate between the instructions of the five selected characters and the remaining 173 candidate characters. This calculation is based on five example instructions, as described in Lines 888–890.
> > >
> > > **Table 3: Word overlap rate between our characters and the remaining 173 characters**
> > >
> > > | **Instruction** | **1** | **2** | **3** | **4** | **5** | **Average** |
> > > |------------------|--------|--------|--------|--------|--------|-----------|
> > > | Mean Overlap     | 0.90   | 0.93   | 0.81   | 0.90   | 0.82   | 0.87      |
> > >
> > > The results in Table 3 demonstrate that our selected characters already cover a broad range of language patterns. Adding more characters would slightly increase the dataset’s scope but is unnecessary for addressing the core problem or establishing a robust benchmark. Notably, the character with the least overlap rate still shares a 65% overlap with our selected characters, highlighting the diminishing returns of including additional characters. For comparison, in the widely used R2R dataset, the evaluation split includes only 11 scans, far fewer than would exist in real-world scenarios. Yet, it is broadly accepted as a standard benchmark for evaluating navigation capabilities. Similarly, our choice of five characters maintains practicality while ensuring meaningful evaluation.
> > >
> > > Moreover, our dataset includes Scene instructions, which encompass styles influenced by professional language and roles, further broadening the diversity of speaking styles. This additional dimension ensures that the five selected User instruction styles are adequate for evaluating adaptation methods.

---

> > > ### Author Response · Authors · 2024-11-24
> > > **Response to Reviewer J6Xq - Part 5**
> > >
> > > ### **4.3 Why These Characters?**
> > > We selected these five characters based on their highly distinct speaking styles, ensuring diversity in age, gender, and language patterns, as described in Lines 890–893.
> > > To measure the diversity of speaking styles, we generated rephrased instructions for each character using the same five Basic instructions and calculated the word overlap rate. Lower overlap rates indicate more distinct language patterns. These results, along with the characters' rankings among all characters in SummScreen, are presented in Table 4.
> > >
> > > **Table 4: Word overlap rate and ranking of four selected characters in GSA-R2R among all characters in SummScreen**
> > >
> > > | **Character** | Keith  | Moira  | Rachel | Sheldon |
> > > |---------------|--------|--------|--------|---------|
> > > | **Overlap Rate** | 0.44   | 0.28   | 0.42   | 0.30    |
> > > | **Ranking**      | 8th    | 1st    | 6th    | 2nd     |
> > >
> > > As stated in Line 893, the child-speaking style was simulated due to the absence of child characters in the SummScreen dataset. This ensures coverage of a general child-speaking style alongside other diverse adult styles.
> > > The overlap rates and rankings confirm that the chosen characters exhibit distinct and diverse language patterns, making them ideal representatives for User instructions. This diversity ensures that our dataset effectively challenges VLN models to adapt to different speaking styles.
> > >
> > > ### **4.4 Future Directions and Reviewer Guidance**
> > >
> > > While we believe our selection is sufficient for the study's aims, we are open to incorporating additional characters in future work. To make this process more systematic, we would appreciate guidance from the reviewer:
> > >
> > > 1. **Objective Criteria**: What kind of experiments or metrics could objectively evaluate the adequacy of selected styles?
> > > 2. **Sample Size**: How many characters would be considered sufficient to demonstrate the generalizability of adaptation methods?
> > >
> > > While we focused on five characters in this study, the flexibility of our method allows for easy expansion to more styles. We plan to release all related code and encourage the community to explore additional styles and evaluate their impact. This collaborative approach will further refine our understanding of instruction adaptation across diverse user personas.

---

> > > > ### Comment · Reviewer_J6Xq · 2024-11-25
> > > >
> > > > Thanks for the authors' replies, I will read them carefully.
> > > > I will think carefully about whether or not to revise my score.

---

> ### Author Response · Authors · 2024-11-22
> **Response to Reviewer J6Xq - Part 2**
>
> ### **2.2 Why We Selected Five Characters**
>
> For this study, we selected five characters with highly distinct speaking styles, ensuring diversity in age, gender, and language patterns. The reasons for this choice are as follows:
>
> 1. **Representativeness**: The five selected characters represent a wide range of user styles, enabling us to effectively evaluate whether adaptation methods can handle significant variations in user instructions.
> 2. **Practicality**:  Adding more characters would not necessarily provide additional insights into the core problem, as the existing selection already covers distinct and challenging styles. We believe that the results on these five characters are sufficiently representative to showcase the instruction adaptation abilities of different methods.
>
> ### **2.3 Generating Infinite Styles**
>
> We appreciate the suggestion of generating instructions in an infinite number of ways, which aligns with our initial considerations. To this end, we have explored persona-based methods, such as the one proposed in [1], which can generate up to **1 billion personas**. However, as described in Lines 856–863, these methods typically provide brief background descriptions of personas without detailed dialog histories. This often leads to generated instructions containing irrelevant or unrealistic content. In contrast, our approach uses detailed dialog profiles as prompts, ensuring that the generated instructions are both realistic and contextually appropriate. Our method prioritizes instruction quality and usability over quantity.
>
> ### **2.4 Code Release**
> We will release all the code for our instruction generation pipeline. This includes the character-based method as presented in our work, and the persona-based method we explored, enabling further exploration of diverse instruction styles. By making this code available, researchers can generate a broader variety of character styles and customize instructions to suit their specific needs and applications.
>
> We agree that expanding our work to include even more diverse instruction styles would be a valuable direction for future research. We are excited to explore such possibilities and thank you for the constructive suggestion.
>
> &nbsp;
>
> [1] Ge, Tao, et al. "Scaling synthetic data creation with 1,000,000,000 personas." arXiv preprint arXiv:2406.20094 (2024).
>
> ---
>
> > ### Weakness 3. GR-DUET Details
>
> We apologize for the lack of sufficient details regarding the baseline DUET model, which may have caused difficulties for readers unfamiliar with this method. While Section 4.1 provides a brief overview of DUET’s key ideas, we acknowledge that this explanation may not be detailed enough to fully convey DUET’s architecture and its integration into our GR-DUET method. To address this, we have added a comprehensive description of DUET in Section A.1.1 of the revised manuscript, including:
>
> - **Model architecture**: Key components and design principles.
> - **Functionality**: How DUET processes textual and visual information individually and then combine them for cross-modal reasoning.
> - **Relevance to GR-DUET**: How DUET forms the foundation of our approach and is adapted within GR-DUET for enhanced performance.
>
> We believe these additions will make the manuscript more accessible to readers unfamiliar with DUET, ensuring they have the necessary background to fully understand our contributions. Thank you for pointing out this gap and giving us the opportunity to improve the clarity of our paper.
>
> ---
>
> > ### Question 1. Human Study Details
>
> Thank you for raising this point. We provide the following clarifications regarding the participants and methodology of our human study:
>
> ### **Participant Demographics**
>
> - The study included 15 participants, comprising university students and staff aged between 20 and 35 years old.
> - The participants represented a diverse range of genders and backgrounds, ensuring a variety of perspectives while maintaining a degree of homogeneity necessary for unbiased evaluations.
>
> ### **Competence of Participants**
>
> The task given to participants was straightforward and did not require specialized knowledge:
>
> - Task 1: Determine whether an instruction aligns with the corresponding trajectory.
>
> - Task 2: Assess whether the instruction exhibits a distinct speaking style.
>
> Given the simplicity of these tasks, we are confident that our participants were competent to perform the evaluation accurately and reliably.
>
> ### **Visualizations**
>
> In Lines 1073–1079 of the manuscript, we include visualizations of the GSA-R2R dataset, which were also shown to the participants during the evaluation. Since OpenReview does not support images in the response, to further ensure transparency, examples of the user interface used in the human study are now included in Section A.8 of the revised manuscript.

---

> ### Author Response · Authors · 2024-11-24
> **Response to Reviewer J6Xq - Part 2**
>
> > ### Question 2: Why LLM Fails
>
> Thank you for raising this question. Regarding the statement, "since the dataset is generated by LLM, why doesn't LLM perform well on it?", we would like to clarify and address this potential misunderstanding.
>
> ### **2.1 Our Dataset is Not Entirely Generated by LLM**
> The GSA-R2R dataset includes novel environments and diverse instructions. While part of the instructions are generated with the help of LLMs, the generation process is rooted in speaker-generated basic instructions. The LLMs used in our pipeline are not reasoning from scratch but transforming existing instructions into different styles based on user-specific prompts and context. Thus, the dataset is not purely LLM-generated and represents a broader mix of styles and real-world challenges.
>
> ### **2.2 Instruction Following vs. Instruction Generation**
> The observed underperformance of LLMs on this dataset stems from the fundamental difference between instruction transformation and instruction-following navigation. Transforming instructions into different styles relies primarily on understanding textual characteristics, such as user preferences and professional speaking habits—a task well-suited to LLMs. In contrast, instruction-following navigation involves multi-modal understanding and reasoning, such as interpreting spatial contexts, visual inputs, and sequential tasks, which are significantly more complex and have not been solved by current LLM-based VLN methods, including InstructNav. This distinction highlights the unique challenges posed by the GSA-VLN task, which cannot be addressed by instruction transformation alone. Moreover, our dataset includes not only the instruction adaptation problem but also the environment adaptation, which is not addressed by LLM-based methods.
>
> ### **2.3 Can LLM solve the instruction adaptation problem?**
> If we only consider the instruction adaptation problem, we would like to provide an additional experiment which has been included in the revised manuscript here. We tested an intuitive idea of whether an LLM could translate styled instructions back into a basic style to facilitate understanding for navigation models. This was evaluated on the Val-R-Scene split using three sets of instructions:
>
>   1. **Basic:** Instructions after Stage 2.
>   2. **Scene:** Instructions transformed from Basic after Stage 3.
>   3. **Translated:** Scene instructions translated back into Basic style by an LLM.
>
>   The performance of these instruction types is summarized in Table 1.
>
> **Table 1: Performance comparison of instruction styles on the Val-R-Scene split.**
>
> | **Instructions** | **Basic** | **Scene** | **Translated** |
> | ---------------- | --------- | --------- | -------------- |
> | SR               | 46.37     | 42.30     | 44.83          |
>
> LLM-based translation improved performance over Scene instructions but did not fully close the gap with Basic instructions. This limitation arises from the open-vocabulary nature of LLMs, which introduces noise and leads to information loss, thereby reducing the effectiveness of the approach. Since this solution is not environment-specific, it falls outside the scope of scene adaptation and is not included in our work.
>
> As noted in our response to Question 1, we have explained the reasons for not including InstructNav in our evaluations and instead adopted MapGPT as a comparable substitute. We believe this addresses any concerns regarding the lack of comparison. We hope this clarification resolves any misunderstandings regarding our dataset and the performance of LLM-based methods. Thank you for the opportunity to elaborate further.

---

> ### Comment · Reviewer_J6Xq · 2024-11-25
>
> Thank the authors for their response. After a careful reading of their data based rebuttals, my concerns are addressed. I kindly suggest that the authors add these analyses to the appendix.
>
> I'll raise my score to 6.

---

> > ### Author Response · Authors · 2024-11-25
> > **Response to Reviewer J6Xq**
> >
> > We sincerely thank you for taking the time to carefully read our rebuttal and for considering our responses. We greatly appreciate your thoughtful engagement with our work and are delighted to hear that your concerns have been addressed.
> >
> > Your feedback and constructive suggestions have been invaluable in improving our manuscript, and we are encouraged by your recognition of our efforts. Thank you once again for your support and for raising your score.

---

### Official Review · Reviewer_WxGN · 2024-11-02

**Soundness:** 2
**Presentation:** 3
**Contribution:** 2
**Rating:** 6
**Confidence:** 4

**Summary:**

This paper presents a task that requires the VLN agent to execute VLN task in only one environment while storing its historical information at the same time. To make the initial parameters of the agent more general, the authors generate more environments and more instructions by using LLM. Finally, the paper also provides some experiment results according to their proposed metrics to further highlight the efficacy of the proposed methods.

**Strengths:**

1、This paper is written with details and clear presentations, easy to follow.

2、The author solves the VLN problem from a new perspective and divides the scenarios into Residential and Non-Residential.

**Weaknesses:**

1、	The novelty of this paper is limited. The GSA-VLN TASK proposed in the paper is still a standard VLN task. The so-called “standard VLN task” mentioned by the paper also includes fine-tuning based on historical information and trained models, which are claimed as the novelty of GSA-VLN in Section 3.2.

2、	Following the previous comment, the GSA-R2R DATASET proposed in the paper uses more environments (HM3D), and then uses tools such as LLM to refine the dataset's quality, which has been a common practice in VLN. Also, the author should not choose to ignore the existing works (e.g., HM3D-AutoVLN[1], Scale VLN[2], YouTube-VLN[3]) have also expanded and refined the VLN dataset when comparing (Table 1). I recommend the authors compare the datasets mentioned above and include them in the main manuscript (e.g. in Table I).

[1] Learning from Unlabeled 3D Environments for Vision-and-Language Navigation

[2] Scaling Data Generation in Vision-and-Language Navigation

[3] Learning Vision-and-Language Navigation from YouTube Videos

3、The comparison metrics in the experimental part are all newly proposed by the authors, which cannot correctly reflect the effectiveness of the method proposed. I suggest that the authors conduct experimental comparisons in standard VLN using common VLN metrics, and compare them on other VLN datasets besides R2R, such as REVERIR, RxR, CVDN and SOON.

**Questions:**

Line 283-286, how is the navigation model used here implemented? How can we ensure that it is a good instruction discriminator? I am concerned that if the navigation model is not trained well, it will not be enough to explain the quality of the instruction.

---

> ### Author Response · Authors · 2024-11-22
> **Response to Reviewer WxGN - Part 1**
>
> We appreciate Reviewer  WxGN for the time and effort in reviewing our paper and offering constructive feedback. Please find our responses to the comments below.
>
> ---
>
> > ### Weakness 1. Novelty of GSA-VLN
>
> We are sorry that reviewer WxGN misunderstood and overlooked the novelty of our task. We would like to clarify that the novelty of GSA-VLN is not derived from "*fine-tuning on historical information or using trained models*". Instead, our work introduces fundamental differences that distinguish GSA-VLN from the standard VLN task, as outlined below:
>
> **Key Differences from Standard VLN Tasks**
>
> 1. **Lifelong, Cumulative Memory**: In standard VLN tasks, agents rely solely on isolated, episode-level history. In contrast, GSA-VLN leverages a lifelong, cumulative memory through the memory bank, enabling agents to continuously adapt to their environment over time. This feature is crucial for handling persistent environments where repeated interactions are required.
> 2. **Dynamic Model Updates**: Unlike standard VLN tasks, where models are fixed during inference, GSA-VLN allows for dynamic model updates at inference time. This enables agents to refine their performance based on scene-specific contexts, facilitating adaptation that is not possible in current VLN paradigms.
>
> These distinctions are clearly outlined in the Introduction section and elaborated upon in Section 3.2 (Lines 189–205), where we provide detailed equations and descriptions to demonstrate how GSA-VLN supports dynamic adaptation to persistent environments, a concept not explored in traditional VLN tasks.
>
> Additionally, our novelty extends beyond proposing the GSA-VLN task. We introduce the GSA-R2R dataset, which significantly expands scene diversity and includes various instruction styles, providing a robust benchmark for evaluating scene-specific adaptation. We also propose the novel GR-DUET method, which achieves state-of-the-art performance. We believe the combination of the GSA-VLN task, the GSA-R2R dataset, and the GR-DUET method constitutes a significant and novel contribution to advancing VLN research. We have clarified these points in the revised manuscript and hope this addresses your concerns regarding the novelty of our work.
>
> ---
>
> > ### Weakness 2. Dataset Comparisons
>
> Thank you for your feedback. Here we would like to clarify that we do not overlook prior works that incorporate additional scenes, and we have discussed the differences between these works and ours in the paper.
>
> As stated in Line 85, we reference HM3D-AutoVLN and ScaleVLN you mentioned, which use the HM3D dataset as augmented **training** data but do not modify the evaluation splits. The Youtube-VLN also belongs to this line of work. However, this contrasts with our GSA-R2R dataset, which preserves the original R2R training data while introducing expanded **evaluation** splits to better assess adaptability in diverse scenarios.
>
> Because of this motivation, the comparison in Table 1 focuses specifically on the evaluation splits of embodied navigation tasks. Since the aforementioned works share the same evaluation splits as R2R, including them in Table 1 would not provide meaningful insights or distinctions. Instead, our dataset comparison highlights the unique features of GSA-R2R, such as its broader range of environments and diverse instruction types, which are absent in prior datasets.
>
> Nevertheless, we acknowledge the need for more detailed discussions to emphasize the differences between GSA-R2R and prior works. In the revised manuscript, we have expanded the *Related Work* section to provide a clearer and more thorough comparison with HM3D-AutoVLN, ScaleVLN, and YouTube-VLN.
>
> ---
>
> > ### Weakness 3. Use of Metrics
>
> We appreciate your feedback and would like to address the concerns regarding the metrics used in our evaluation and the datasets chosen for experimentation.
>
> ### **3.1 Clarification on Evaluation Metrics**
>
> We believe there may be a misunderstanding regarding the metrics used in our evaluation. The metrics we use in our evaluation, including Trajectory Length (TL), Navigation Error (NE), Success Rate (SR), Success weighted by Path Length (SPL), and normalized Dynamic Time Warping (nDTW), are standard and widely recognized in VLN research. These metrics have been consistently used as benchmarks in prior works to comprehensively evaluate navigation success, efficiency, and instruction fidelity [1, 2]. **None of these metrics are newly proposed by us**. These metrics provide a comprehensive evaluation of agent performance and are sufficient to demonstrate the effectiveness of our method. We believe that adhering to these standard metrics ensures the comparability and validity of our results within the VLN research community.

---

> ### Author Response · Authors · 2024-11-22
> **Response to Reviewer WxGN - Part 2**
>
> ### **3.2 Comparison to Other Datasets**
>
> Our primary contributions focus on introducing the GSA-VLN task and the GSA-R2R dataset, which are specifically designed to enable and evaluate **scene-specific adaptation**. Therefore, our experiments are centered around these contributions to provide meaningful insights into the proposed task and dataset. The datasets you mentioned, such as REVERIE, RxR, CVDN, and SOON, belong to traditional VLN settings and do not encompass the scene and instruction diversity emphasized by GSA-VLN. While these datasets provide valuable benchmarks, they lack the specific characteristics necessary for evaluating scene-specific adaptation, which is the core focus of our work.
>
> We acknowledge that adapting these datasets into GSA-VLN (e.g., **GSA-REVERIE** or **GSA-CVDN**) would be an exciting direction for future research. However, such extensions are beyond the scope of this paper. We appreciate your suggestion and will consider these opportunities in our future work.
>
> &nbsp;
>
> [1] Wang, Zun, et al. "Scaling data generation in vision-and-language navigation." ICCV. 2023.
>
> [2] Liu, Rui, et al. "Volumetric Environment Representation for Vision-Language Navigation." CVPR. 2024.
>
> ---
>
> > ### Question 1. Instruction Quality
>
> ### **1.1 How is the navigation model used here implemented?**
>
> The implementation details of our model-based selection process are provided in Section A.2 (Lines 832–838) of the paper. In summary, we utilize a DUET model trained on unselected paths from the same 150 environments chosen for the GSA-R2R dataset. This setup ensures that:
>
> - The GSA-R2R instructions serve as a **validation seen split** for the model
> - The model is familiar with the environment's visual and spatial characteristics but has not encountered the specific instructions.
>
> This design isolates the evaluation from the impact of unfamiliar environments, allowing the model to focus solely on instruction quality. Using this approach, the model achieves a high Success Rate (SR) of **73.6%**, indicating robust performance.
>
> ### **1.2 How can we ensure that it is a good instruction discriminator?**
>
> We designed this process to identify the most robust model for evaluating instruction quality. The underlying rationale is:
>
> 1. If a well-trained model can successfully navigate to the correct target using a specific instruction, other sufficiently trained models should also have the potential to succeed.
> 2. Conversely, it is highly unlikely for a model to consistently reach the correct target when following incorrect instructions.
>
> This approach validates the model’s role as a reliable instruction discriminator.
>
> Furthermore, the baseline adaptation models, including GR-DUET, share the same architecture as DUET. This ensures that they only need to bridge the gap from **unseen to seen environments** to perform well with the provided instructions. This aligns directly with the goal of scene adaptation, as outlined in Line 48 of the paper.
>
> ### **1.3 Is it enough to explain the quality of the instruction?**
>
> To ensure comprehensive validation, we supplement the model-based evaluation with human evaluation, as detailed in Section 3.3.4. Table 2 reports that GSA-R2R instructions achieve approximately **80% match accuracy**, closely aligning with the 86% human success rate on the original R2R dataset. This similarity highlights the high quality of the GSA-R2R instructions, demonstrating their alignment with human expectations. These results, combined with the model-based evaluation, provide strong evidence of the robustness and quality of the instructions in GSA-R2R.
>
> We hope this clarifies the robustness of our approach and demonstrates that our model effectively discriminates between instruction quality.

---

> ### Author Response · Authors · 2024-11-25
> **Kind Request for Discussion and Reconsideration of the Score**
>
> We thank Reviewer WxGN for the valuable feedback and insightful suggestions, which have helped us refine and clarify our work. We have carefully addressed all the raised concerns in our response.
>
> We would greatly appreciate it if Reviewer WxGN could provide further feedback. Your input is invaluable to ensuring the quality and clarity of our work. Or, if our responses have satisfactorily resolved the concerns, we respectfully request reconsideration of the score based on the clarifications and improvements provided.

---

> > ### Comment · Reviewer_WxGN · 2024-11-25
> >
> > I appreciate the author's responses. The authors address most of my concerns. I decide to change my score to acceptance. I hope the authors revise the paper according to the reviewers' reviews.

---

> > > ### Author Response · Authors · 2024-11-25
> > > **Response to Reviewer WxGN**
> > >
> > > Thank you for your thoughtful review and for taking the time to carefully consider our responses. We are pleased to hear that we addressed most of your concerns.
> > >
> > > We have carefully revised the manuscript based on the valuable feedback provided by you and the other reviewers. The updated version reflects these revisions, including improvements to clarity, additional analyses, and expanded discussions. We are committed to further refining our work and welcome any additional suggestions to enhance its quality.
> > >
> > > Your constructive feedback has been invaluable in strengthening our paper, and we deeply appreciate your recommendation. Thank you once again for your insights and support!

---

### Official Review · Reviewer_t9oJ · 2024-11-03

**Soundness:** 3
**Presentation:** 3
**Contribution:** 4
**Rating:** 8
**Confidence:** 4

**Summary:**

This paper presents General Scene Adaptation for Vision-and-Language Navigation (GSA-VLN), a new VLN task where agents adapt to and improve in a specific environment over time, making it closer to real-world applications. To support this, the authors introduce GSA-R2R, a dataset that expands on Room-to-Room (R2R) by adding more diverse environments and instruction styles, including out-of-distribution examples. They also propose Graph-Retained DUET (GR-DUET), a method that uses memory-based navigation graphs and scene-specific training to help agents learn and retain scene-specific information, achieving strong results on the GSA-R2R benchmarks.

**Strengths:**

1. This paper introduces the novel General Scene Adaptation for Vision-and-Language Navigation (GSA-VLN) task, filling a critical gap in VLN research by focusing on adaptation in persistent environments. Rather than assuming agents will encounter only unseen environments, GSA-VLN models a more realistic scenario where agents learn and improve over time within a familiar setting. This shift in task formulation is both timely and innovative, especially as VLN moves toward practical applications.
2. The paper demonstrates rigorous methodology in creating the GSA-R2R dataset, expanding on the Room-to-Room (R2R) dataset with a variety of environments, instruction styles, and out-of-distribution examples to thoroughly test agent adaptability. The proposed Graph-Retained DUET (GR-DUET) model is well-designed, combining memory-based navigation graphs with a scene-specific training strategy, and shows significant performance improvements across metrics.
3. The paper is clearly organized and effectively conveys the importance of long-term scene adaptation in VLN.

**Weaknesses:**

1. The GR-DUET method involves a memory bank and a global graph that retains historical information across episodes. As the memory and graph size increase, the model’s computational requirements may grow significantly, particularly for long-term navigation in large environments. While the paper includes an environment-specific training strategy to limit graph expansion, providing an analysis of computational costs and potential trade-offs between memory retention and scalability would strengthen the model's practicality for deployment on resource-constrained systems.
2. While the GSA-R2R dataset is a notable improvement over existing datasets for testing scene-specific adaptation, it may still fall short in representing the full diversity of real-world environments and interaction styles. The dataset includes a mix of residential and non-residential scenes, but further validation with a broader set of real-world environments could strengthen the model's applicability. Including additional scene types, such as commercial or outdoor spaces, or testing in dynamic environments where the layout changes over time, would push the dataset closer to real-world settings.
3. Although the paper’s three-stage instruction generation pipeline enhances instruction diversity, more detailed analysis on how different instruction styles (e.g., Basic, User, Scene) impact agent performance would be valuable. For instance, specific ablation studies on each instruction type could clarify how robust the GR-DUET model is to variances in language, phrasing, and style. Additionally, investigating how the model generalizes across speakers with different dialects or levels of detail in instructions could provide actionable insights into improving instruction handling.

**Questions:**

Please refer to the weaknesses section.

---

> ### Author Response · Authors · 2024-11-22
> **Response to Reviewer t9oJ - Part 1**
>
> We are grateful to Reviewer t9oJ for dedicating time and effort to reviewing our paper and providing thoughtful and constructive feedback. Our detailed responses to the comments are outlined below.
>
> ---
>
> >### Weakness 1. Scalability and Resource Use
>
> Although the memory bank in GR-DUET updates continuously during navigation, it stabilizes once the agent has explored most of the environment. Updates after this point are minimal, involving only new instructions and actions, which require relatively little memory. Moreover, GR-DUET employs coarse-grained embeddings for nodes that are not neighbors of the current node, ensuring that GPU memory usage does not grow significantly even when processing the entire graph. To illustrate this, we present key computational metrics across episodes for one of the largest environments in GSA-R2R in Table 1.
>
> **Table 1: Variations in computational costs of GR-DUET across different episodes.**
>
> | Episode             | 1    | 100  | 200  | 300  | 400  | 500  | 600  |
> | ------------------- | ---- | ---- | ---- | ---- | ---- | ---- | ---- |
> | Graph Coverage (%)  | 4.1  | 68.1 | 81.1 | 89.6 | 94.1 | 97.4 | 98.5 |
> | GPU memory (MB)     | 823  | 2917 | 3609 | 3609 | 3609 | 3609 | 3609 |
> | CPU memory (MB)     | 5174 | 5252 | 5278 | 5284 | 5290 | 5291 | 5291 |
> | Inference Time (ms) | 12   | 56   | 63   | 65   | 66   | 66   | 67   |
>
> As agents execute more instructions, we observe gradual increases in CPU memory usage, GPU memory usage, and inference time. However, these metrics stabilize as the graph coverage approaches 1, indicating that the agent has explored most of the environment.
>
> From the model perspective, GR-DUET contains 180 million parameters, occupying only 2.1 GB of disk space. The maximum inference time of 67 ms translates to a throughput of 15 frames per second, making it suitable for real-time navigation tasks. Its computational complexity is 1.63 GFLOPs (excluding visual feature extraction), enabling deployment on robots without excessive computational demands. **All these statistics demonstrate that GR-DUET is computationally scalable and practical for deployment on resource-constrained systems. .** In the revised manuscript (Section A.5), we include additional visualizations and analyses for two more environments of varying sizes to further validate these findings.
>
> ---
>
> >### Weakness 2. Dataset Diversity
>
> Thank you for your insightful suggestion. To minimize the sim-to-real gap, current VLN research relies on photorealistic environments, which limits the datasets available for studying VLN compared to navigation tasks that can utilize synthetic or procedurally generated datasets. Within these constraints, we have made significant efforts to expand the diversity of GSA-R2R to include **20 distinct scene types**, compared to just six in R2R. This diversity covers a wide range of daily scenarios and exceeds that of existing embodied navigation datasets, as highlighted in Table 1 of our paper.
>
> Regarding the three types of scenes recommended:
>
> - Commercial Spaces: We already include multiple commercial spaces such as cinemas, shops, and restaurants, as illustrated in Figure 2 of our paper.
> - Outdoor Spaces: While the dataset focuses on indoor environments, some scenes include outdoor elements such as gardens, yards, and swimming pools adjacent to houses. However, outdoor navigation tasks [1-2] require fundamentally different capabilities and face challenges distinct from indoor navigation. Consequently, outdoor VLN is generally studied separately within the research community. Since GSA-R2R focuses on indoor, scene-specific adaptation, incorporating full outdoor spaces falls outside the scope of this work.
> - Dynamically Environments: Dynamically changing environments are a valuable and realistic direction for embodied navigation research [3-4]. However, expanding GSA-R2R to include dynamic environments introduces significant challenges, such as environment manipulation and maintaining consistency in dynamic layouts. These challenges extend beyond the current scope of our work. We believe that future research can build on the foundation provided by GSA-R2R to address dynamic adaptation.
>
> We agree that extending GSA-R2R to represent even broader real-world scenarios, including additional scene types or dynamic environments, would further strengthen its applicability. We appreciate the reviewer’s valuable suggestions and will consider these directions as future works.
>
> &nbsp;
>
> [1] Liu, Shubo, et al. "Aerialvln: Vision-and-language navigation for UAVs." ICCV. 2023.
>
> [2] Li, Jialu, et al. "Vln-video: Utilizing driving videos for outdoor vision-and-language navigation." AAAI. 2024.
>
> [3] Li, Heng, et al. "Human-aware vision-and-language navigation: Bridging simulation to reality with dynamic human interactions." NeurIPS. 2024.
>
> [4] Zhou, Qinhong, et al. "HAZARD Challenge: Embodied Decision Making in Dynamically Changing Environments." ICLR. 2024.

---

> ### Author Response · Authors · 2024-11-22
> **Response to Reviewer t9oJ - Part 2**
>
> > ### Weakness 3. Instruction Adaptation
>
> Thank you for your insightful suggestions. We are happy to provide more details about our findings and insights on the instruction adaptation problem.
>
> ### **3.1 Ablation Study on Each Instruction Type**
> We have already included an ablation study comparing Basic, Scene, and User instructions in Table 11 of the paper, keeping environments and paths constant. The results show that stylized instructions (User and Scene) lead to a similar SR drop for the DUET model. This demonstrates that these styles introduce additional challenges due to their increased complexity and variability in language.
>
> ### **3.2 Further Analysis**
> Our primary focus has been on establishing a high-quality benchmark for studying instruction adaptation. While GR-DUET primarily addresses environment-side adaptation without optimizing for instruction styles, we conducted extensive experiments to evaluate how existing methods handle this challenge, including those with potential for instruction adaptation, such as Masked Language Modeling (MLM) and Back-Translation (BT).
>
> Table 5 in the paper presents results for different adaptation methods across varying instruction styles. TTA methods, such as TENT, perform better with Scene instructions due to the distinct language patterns introduced by conversational fillers, which are more recognizable than the subtle word variations in User instructions. However, the advantage brought by Back-Translation (BT) is significantly reduced in instructions with diverse styles compared to Basic instructions. This is because BT struggles with larger gaps between the training corpus and human instructions, highlighting its difficulty in adapting to speaker-specific variations effectively.
>
> We further present the mean performance and standard deviation (std) of various methods across different speakers in User instructions in Table 2.
>
> **Table 2: Mean SR and Standard Deviation (std) of baseline methods across different speakers.**
>
> | Method | DUET  | +MLM  | +MRC  | +BT   | +TENT | +SAR  | GR-DUET |
> | ------ | ----- | ----- | ----- | ----- | ----- | ----- | ------- |
> | Mean   | 54.58 | 55.20 | 54.48 | 59.04 | 53.88 | 51.72 | 64.76   |
> | Std    | 1.46  | 1.18  | 1.41  | 1.93  | 1.36  | 1.44  | 1.88    |
>
> MLM achieves the lowest std, demonstrating improved adaptation to instruction styles by learning speaker-specific vocabulary. BT achieves the highest overall performance among adaptation-based methods but also shows the highest std, reflecting its sensitivity to the training corpus of the speaker model it uses. Specifically, BT overfits to its training speaker's style, leading to inconsistencies when applied to diverse styles, as it amplifies performance variations.
>
> These results underscore the challenges of instruction adaptation and provide a foundation for future research.
>
> ### **3.3 Different Dialects or Detail Levels**
> We appreciate your suggestion to explore the impact of dialects and detail levels. our approach models speaking styles as a general term including differences in vocabulary, catchphrases, dialects, and detail levels. For example, the characters have backgrounds from different places, naturally introducing dialects into their instructions. This diversity mirrors real-world scenarios and provides a more comprehensive test of instruction adaptation by including linguistic variability. Rather than isolating specific variables like dialects, our general setup provides a robust framework for evaluating model performance across a wide range of language styles. Models capable of adapting to these combined variables are better suited to real-world applications. Future work could delve into isolating individual factors to further analyze their specific impact and propose tailored methods for improving adaptability.
>
>
> We hope our findings inspire further exploration of model-side adaptations for specific language patterns in VLN tasks. Instruction style adaptation remains a promising direction for future research, and we look forward to seeing the community build upon our benchmark and methods.

---

### Official Review · Reviewer_dHDF · 2024-11-04

**Soundness:** 3
**Presentation:** 3
**Contribution:** 2
**Rating:** 6
**Confidence:** 4

**Summary:**

The paper proposes GSA-VLN (General Scene Adaptation for Vision-and-Language Navigation), a task designed to enhance the performance of navigation agents by enabling them to adapt to specific environments, particularly when exploring in the same environment over an extended period. The authors also introduce GSA-R2R, an expanded version of the HM3D and MP3D dataset, offering richer environments and more diverse instructions. Additionally, they present a novel method, GR-DUET, which improves navigation performance by utilizing memory mechanisms and updating graph structures.

**Strengths:**

Novelty:
The paper introduces a new task, GSA-VLN, which focuses on the long-term adaptation of agents within specific environments, a capability with significant potential for real-world applications.

Dataset Contribution:
The authors present the GSA-R2R dataset, which extends the existing R2R dataset by using GPT-4 and a three-stage method to generate instructions in various speaking styles. The dataset is divided into residential and non-residential environments, serving as in-distribution (ID) and out-of-distribution (OOD) data, respectively.

Method Design:
The GR-DUET method integrates topological graphs with memory mechanisms, effectively preserving historical information and updating it continuously during navigation. This approach demonstrates notable improvements in performance, particularly in OOD (non-residential) scenarios.

Experimental Results:
The paper compares GR-DUET with optimization-based and memory-based methods across different environment and speaking style splits. The experiments highlight the feasibility of the GSA-VLN task and the effectiveness of the GR-DUET method in various settings.

**Weaknesses:**

Please see the Questions section for detailed improvement suggestions and questions.
I look forward to the authors' responses to these questions, as addressing these points could significantly clarify some of the paper's contributions and limitations. I am open to adjusting my score if the authors provide further insights or resolve the concerns raised above.

**Questions:**

1. Memory Mechanism Scalability:
While the memory-based approach in GR-DUET performs well in your experiments, how does this method scale to larger or more complex environments? As the environment size or the number of instructions increases, the memory bank may become too large to manage efficiently. Could you provide further analysis or experiments that demonstrate how the method performs with continuous accumulation of data in larger datasets or more complex environments?

2. the paper lacks a detailed discussion on how the memory is utilized, including how similar tasks are stored, how memory is retrieved and assessed for relevance and validity, and how prior knowledge is leveraged. Is the memory bank pre-set or updated dynamically? If it is updated dynamically, how is the correctness of the stored memories ensured, especially when handling diverse memories? How are the initial model parameters (L194, L198) initialized to ensure sufficient generalization? Please provide more details

3. Furthermore, other memory-based VLN methods, such as SG-Nav [1], provide more detailed storage and query mechanisms based on topological graphs and memory updates. Could you compare your approach with SG-Nav in terms of performance or highlight any differences and advantages?

4. Adaptation to Instruction Styles:
You mention using GPT-4 and a three-stage process to generate different instruction styles, but it remains unclear how the agent adapts to these varying styles over time. Could you provide more quantitative and qualitative results on how GR-DUET handles changes in style, particularly in OOD environments? A deeper analysis of how different speaking styles affect agent performance and adaptability would offer valuable insights into the robustness of your method in real-world scenarios, where user communication patterns may vary significantly.

5. Unsupervised Learning and Adaptation Efficiency:
The paper suggests that agents in GSA-VLN can improve their performance over time using unsupervised learning techniques. Could you clarify how quickly the agents adapt in different environments? Are there any cases where adaptation is less effective or slower? Are there specific environments where the memory mechanism struggles to adapt? A more detailed breakdown of adaptation speed and efficiency across different environment types would help clarify the limitations of your approach and guide future improvements.

6. Practical Deployment and Real-World Use Cases:
The GSA-VLN task is well-motivated by real-world scenarios, but the paper does not provide a detailed discussion on how the proposed method could be deployed in practical systems. Could you elaborate on the computational and memory overhead of your approach in real-time systems, such as those used in robotics or autonomous agents?

Reference:
[1] Yin, Hang, et al. "SG-Nav: Online 3D Scene Graph Prompting for LLM-based Zero-shot Object Navigation." arXiv preprint arXiv:2410.08189 (2024).

---

> ### Author Response · Authors · 2024-11-22
> **Response to Reviewer dHDF - Part 1**
>
> We appreciate Reviewer dHDF for the time and effort in reviewing our paper and offering constructive feedback. Please find our responses to the comments below.
>
> ---
>
> > ### Question 1. Memory Mechanism Scalability
>
> We agree that the scalability of GR-DUET is an important consideration for addressing the scene adaptation problem. Below, we provide further analysis to demonstrate the scalability of our method in larger and more complex environments from both computational and performance perspectives.
>
> ### **1.1 Computational Costs and Memory Usage**
>
> Although the memory bank in GR-DUET updates continuously during navigation, it stabilizes once the agent has explored most of the environment. Updates after this point are minimal, involving only new instructions and actions, which require relatively little memory. Moreover, GR-DUET employs coarse-grained embeddings for nodes that are not neighbors of the current node, limiting GPU memory growth despite inputting the entire graph into the model. To illustrate this, we analyze key computational metrics across episodes for one of the largest environments in GSA-R2R (Table 1).
>
> **Table 1: Variations in computational costs of GR-DUET across different episodes.**
>
> | Episode             |      1      |     100     |     200     |     300     |     400     |     500     |     600     |
> | ------------------- | ----------- | ----------- | ----------- | ----------- | ----------- | ----------- | ----------- |
> | Graph Coverage (%)  |     4.1     |    68.1     |    81.1     |    89.6     |    94.1     |    97.4     |    98.5     |
> | GPU memory (MB)     |     823     |    2917     |    3609     |    3609     |    3609     |    3609     |    3609     |
> | CPU memory (MB)     |    5174     |    5252     |    5278     |    5284     |    5290     |    5291     |    5291     |
> | Inference Time (ms) |     12      |     56      |     63      |     65      |     66      |     66      |     67      |
>
> As agents execute more instructions, we observe gradual increases in CPU memory usage, GPU memory usage, and inference time. However, when the graph coverage approaches 100%, indicating that the agent has explored most places, these metrics stabilize with minimal additional overhead. **This demonstrates that GR-DUET is computationally scalable and does not sacrifice much memory efficiency for improved performance.** In the revised manuscript (Section A.5), we include additional visualizations and analyses for two more environments of varying sizes.
>
> ### **1.2 Navigation Performance**
>
> In large environments, the memory bank grows, introducing more inputs to the model. However, GR-DUET’s pre-training stage on the entire ground-truth graph ensures that it effectively handles these inputs. By focusing on surrounding viewpoints rather than the full graph, GR-DUET mitigates the impact of environment size on computational efficiency and performance. To validate this, we compare the performance gap (Success Rate, SR) between the original DUET and GR-DUET across all 150 environments in GSA-R2R with varying numbers of viewpoints (Table 2).
>
> **Table 2: Mean SR gap between GR-DUET and original DUET by environment size.**
>
> | Viewpoint Number    |     50-100     |     100-150     |     150-200     |     200-250     |     250-300     |     >300      |
> | ------------------- | -------------- | --------------- | --------------- | --------------- | --------------- | ------------- |
> | Mean SR Gap (%)     |      9.07      |      10.17      |      7.57       |      9.14       |      8.02       |     10.74     |
>
>
> The results show that GR-DUET consistently outperforms the original DUET, even in environments with a large number of viewpoints. **This indicates that GR-DUET effectively learns to focus on relevant parts of the graph, ensuring scalability in large environments.** Additionally, we include a scatter plot of SR gaps versus navigable areas in Section A.5 of the revised manuscript, which further confirms that GR-DUET consistently performs better in all large environments.
>
> ---
>
> > ### Question 2. Memory Mechanism Details
>
> The focus of our paper is on introducing a new task and dataset, which is why most of the content is dedicated to detailing these aspects. The method presented serves as a baseline, and due to space constraints, certain details were not included in the original manuscript. We thank the reviewer for highlighting this, and we provide the requested clarifications below, which have also been added to the revised manuscript.
>
> ### **2.1 How is the memory utilized?**
>
> In GSA-VLN, our framework does not impose a specific way to utilize the information in the memory bank. Instead, it provides the entire history of information, enabling different methods to selectively leverage this data to improve performance for improved performance. The memory bank can be used in various ways, such as additional input for decision-making or for unsupervised learning purposes, as outlined in Section 3.2.

---

> ### Author Response · Authors · 2024-11-22
> **Response to Reviewer dHDF - Part 2**
>
> ### **2.2 Is the memory bank pre-set or updated dynamically?**
>
> The dynamic update procedure is detailed in Equation 1 of the paper. As described in Lines 180-188, *"This memory bank dynamically expands as the agent executes instructions, capturing four key components: visual observations ($\mathbf{O}$), the instructions ($X$), selected actions ($\mathbf{A}$), and trajectory paths ($\mathbf{P}$). For example, after executing $k$ instructions in environment $E$, the memory bank is updated as follows:
> $ \mathcal{M}_E = \{X^{1:k}, \mathbf{O}^{1:k}, \mathbf{A}^{1:k}, \mathbf{P}^{1:k}\}$ ."*
>
> After each episode, the data collected during that episode is stored in the memory bank, enabling the agent to build a comprehensive history of its navigation.
>
> ### **2.3 How is the correctness of the stored memories ensured?**
>
> The stored memories represent the agent’s execution history, including user-provided instructions, visual observation from sensors, actions taken, and trajectories followed. Since this data directly reflects the agent’s experience and is ``factual'', the concept of "correctness" does not strictly apply. We understand the concerns that there may exist misalignment between instructions and paths due to navigation errors. However, all the memories are treated as unlabeled data and are primarily used for unsupervised learning techniques, such as Back-Translation or Masked Language Modeling.
>
> For example, in GR-DUET, trajectory and observation data are employed to construct a global, consistent graph for decision-making. These components are directly derived from the agent’s observations and are inherently accurate, ensuring reliable support for the decision-making process.
>
> ### **2.4 How are the initial model parameters (L194, L198) initialized?**
>
> The initial model parameters are determined by the specific method but must ensure sufficient generalization across diverse environments. While the goal is to adapt to environment-specific models at inference, the trained model must retain universality to handle a wide range of environments. Overfitting the model to a specific target environment during training would compromise its generalization ability, making the method unrealistic and impractical. To address this, we train GR-DUET on the full training split of R2R and augment the instructions with data from PREVALENT. This approach ensures that GR-DUET learns generalized navigation capabilities while remaining adaptable during inference.
>
> We have incorporated these details into Section 3.2 of the revised manuscript. Thank you for highlighting this opportunity to enhance the clarity of our work.
>
> ---
>
> > ### Question 3. Comparison with SG-Nav
>
> Thank you for the suggestion. We have carefully read SG-Nav and found its approach to be highly impressive. The idea of constructing consistent 3D scene graphs aligns closely with the goals of our GSA-VLN task and represents a potential solution for enhancing scene adaptation. This concept is also reflected in the OVER-NAV baseline used in our work, which constructs an omnigraph similar to scene graphs to consistently record encountered objects. However, both GR-DUET and OVER-NAV primarily store historical information within a topological map, limiting the stored data to visual observations or detected objects. In contrast, the 3D scene graphs used in SG-Nav offer a more powerful representation of the environment, which could potentially help agents retrieve and leverage relevant historical information more effectively for navigation tasks.
>
> Unfortunately, SG-Nav could not be directly applied to our VLN task due to fundamental differences in task design:
>
> 1. **Task Objective**: SG-Nav is tailored for object-goal navigation, where the input is a target object rather than fine-grained natural language instructions.
> 2. **Action Space**: SG-Nav employs low-level actions (e.g., move forward, turn left, turn right, and stop), whereas VLN requires the agent to select a waypoint from a discrete set of candidate locations based on complex instructions.
> 3. **Adapting SG-Nav to VLN**: Substantial modifications would be required to adapt SG-Nav’s approach to hierarchical VLN tasks, particularly for handling fine-grained instructions and multi-step navigation scenarios.
>
> While we could not include SG-Nav as a direct baseline due to these technical challenges, we plan to explore its ideas, particularly its 3D scene graph representation, in future work. This could lead to improved memory mechanisms and more robust scene adaptation in VLN tasks. We have also included a discussion of SG-Nav in the *Related Work* section of the revised manuscript.
>
> We believe we have made a comprehensive effort to include all relevant VLN works with long-term memory mechanisms as baselines. However, if you have additional baseline suggestions, we would be happy to consider them and incorporate them into our analysis.

---

> ### Author Response · Authors · 2024-11-22
> **Response to Reviewer dHDF - Part 3**
>
> > ### Question 4. Instruction Style Adaptation
>
> Thanks for raising the point about instruction adaptation. We would like to clarify the scope of our work and present our findings and insights on this emerging problem.
>
> ### **4.1 Novelty of Instruction Style Adaptation**
>
> Previous VLN works mainly utilize instructions with plain and concise language, overlooking personal speaking habits. In this work, we are **the first to propose the problem of instruction style adaptation**, addressing this critical gap. While our primary focus is on establishing a benchmark with sufficient high-quality data to study this problem, we acknowledge that solving instruction adaptation requires future efforts. Our GR-DUET model focuses more on environment-side adaptation without specific optimization for instruction styles. This limitation is explicitly discussed in Section A.5 of our paper. We view instruction adaptation as an open problem and a promising research direction enabled by the benchmarks and data provided by our work.
>
> ### **4.2 Analysis of Performance**
>
> Although our method does not include instruction-specific optimizations, we conducted extensive experiments to evaluate how existing methods handle this challenge, including those with potential for instruction adaptation, such as Masked Language Modeling (MLM) and Back-Translation (BT).
>
> Table 5 in the paper presents results for different adaptation methods across varying instruction styles. TTA methods, such as TENT, perform better with Scene instructions due to the distinct language patterns introduced by conversational fillers, which are more recognizable than the subtle word variations in User instructions. However, the advantage brought by Back-Translation (BT) is significantly reduced in instructions with diverse styles compared to Basic instructions. This is because BT struggles with larger gaps between the training corpus and human instructions, highlighting its difficulty in adapting to speaker-specific variations effectively. We further present the mean performance and standard deviation (std) of various methods across different speakers in User instructions in Table 3.
>
> **Table 3: Mean SR and Standard Deviation (std) of baseline methods across different speakers.**
>
> | Method | DUET  | +MLM  | +MRC  | +BT   | +TENT | +SAR  | GR-DUET |
> | ------ | ----- | ----- | ----- | ----- | ----- | ----- | ------- |
> | Mean   | 54.58 | 55.20 | 54.48 | 59.04 | 53.88 | 51.72 | 64.76   |
> | Std    | 1.46  | 1.18  | 1.41  | 1.93  | 1.36  | 1.44  | 1.88    |
>
> MLM achieves the lowest std, demonstrating improved adaptation to instruction styles by learning speaker-specific vocabulary. BT achieves the highest overall performance among adaptation-based methods but also shows the highest std, reflecting its sensitivity to the training corpus of the speaker model it uses. Specifically, BT overfits to its training speaker's style, leading to inconsistencies when applied to diverse styles, as it amplifies performance variations.
>
> These results underscore the challenges of instruction adaptation and provide a foundation for future research.
>
> ### **4.3 Potential Solutions**
>
> We have also explored additional strategies for addressing instruction adaptation:
>
> - **Weighted Masked Language Modeling**: Vanilla MLM treats all words equally and randomly masks a proportion of them, which is inefficient for focusing on key or unseen words. We modified the MLM approach to prioritize masking unseen or rare words specific to a speaker's style. This achieved a 1% improvement in SR, but required extensive hyperparameter tuning for each environment, making it impractical for real-world deployment.
>
> - **LLM-based Instruction Translation**: We tested an intuitive idea of whether an LLM could translate styled instructions back into a basic style to facilitate understanding for navigation models. This was evaluated on the Val-R-Scene split using three sets of instructions:
>
>   1. **Basic:** Instructions after Stage 2.
>   2. **Scene:** Instructions transformed from Basic after Stage 3.
>   3. **Translated:** Scene instructions translated back into Basic style by an LLM.
>
>   The performance of these instruction types is summarized in Table 4.
>
> **Table 4: Performance comparison of instruction styles on the Val-R-Scene split.**
>
> | **Instructions** | **Basic** | **Scene** | **Translated** |
> | ---------------- | --------- | --------- | -------------- |
> | SR               | 46.37     | 42.30     | 44.83          |
>
> LLM-based translation improved performance over Scene instructions but did not fully close the gap with Basic instructions. This limitation arises from the open-vocabulary nature of LLMs, which introduces noise and leads to information loss, thereby reducing the effectiveness of the approach. Since this solution is not environment-specific, it falls outside the scope of scene adaptation and is not included in our work.

---

> ### Author Response · Authors · 2024-11-22
> **Response to Reviewer dHDF - Part 4**
>
> While we have not fully solved the instruction adaptation problem, our work lays the groundwork by:
>
> 1. Establishing a benchmark dataset with diverse instruction styles.
> 2. Providing a comprehensive evaluation of how existing adaptation methods handle this challenge.
>
> We hope our findings will inspire further exploration into model-side adaptations for specific language patterns in VLN tasks. Instruction style adaptation remains an exciting area for future research, and we look forward to seeing the community build upon our dataset and methods.
>
> ---
>
> > ### Question 5. Adaptation Efficiency
>
> Our work mainly focuses on scenarios where the agent operates in a persistent environment over an extended period, potentially its entire lifetime. In such cases, the overall performance after adaptation is more critical than the adaptation speed. This explains why we include 600 paths for each environment in GSA-R2R, significantly more than previous VLN datasets, ensuring sufficient instructions for effective adaptation. Nevertheless, we conducted further analyses on GR-DUET to better understand its adaptation efficiency.
>
> ### **5.1 How quickly do the agents adapt to different environments?**
>
> GR-DUET builds a global graph for adaptation, which stabilizes once the agent has explored most parts of the environment. Based on the graph coverage data (Table 1 of Question 1), we find that 90% coverage is achieved with at most 400 instruction-trajectory pairs in large environments and as few as 100 pairs in small environments. To measure adaptation speed, we treat the first *X* instructions (the number required to reach 90% graph coverage) as the adaptation phase and divide them into groups of 50 instructions. Performance within each group is measured, and linear regression is applied to calculate the slope of performance improvement, serving as a proxy for adaptation speed. Results show that among the 150 scans, 94 achieved a positive slope, with a mean slope of **0.26**, indicating rapid adaptation in most cases.
>
> ### **5.2 Are there any cases where adaptation is less effective or slower?**
>
> To understand slower or less effective adaptation, we analyzed adaptation speed across various environmental characteristics.
>
> First, Table 5 shows the mean adaptation slopes in environments with different numbers of floors.
>
> **Table 5: The adaptation speeds of GR-DUET in environments with different number of floors.**
>
> | Floor Number | 1    | 2    | 3    | 4    |
> | ------------ | ---- | ---- | ---- | ---- |
> | Mean Slopes  | 0.24 | 0.19 | 0.05 | 0.40 |
>
> Adaptation becomes less effective as the number of floors increases, except for a few cases with four floors. This is intuitive, as distinct floor layouts and styles make prior memory from other floors less relevant to the current navigation.
>
> Conversely, Table 6 shows that adaptation efficiency improves as the number of rooms increases, particularly in buildings with more than 15 rooms.
>
> **Table 6: The adaptation speeds of GR-DUET in environments with different number of rooms.**
>
> | Room Number | 1-5  | 5-10 | 10-15 | 15-20 | 20-25 | 20-25 | >25  |
> | ----------- | ---- | ---- | ----- | ----- | ----- | ----- | ---- |
> | Mean Slopes | 0.48 | 0.04 | 0.06  | 0.20  | 0.28  | 0.36  | 0.70 |
>
> After viewing specific buildings, we find that environments with many rooms (e.g., hotels or student accommodations) often have repetitive layouts and identical rooms, allowing the agent to leverage memory from similar spaces effectively. These findings suggest that **GR-DUET performs well in environments with repetitive structures but struggles in environments with dissimilar memory** (e.g., multi-floor buildings with distinct layouts).
>
> ### **5.3 Are there specific environments where the memory mechanism struggles to adapt?**
>
> We calculated mean adaptation slopes for different scene and instruction types, as shown in Table 7.
>
> **Table 7: The adaptation speeds of GR-DUET in different types of scenes and instructions.**
>
> | Type        | Residential | Non-residential | Basic Inst. | Scene Inst. | User Inst. |
> | ----------- | ----------- | --------------- | ----------- | ----------- | ---------- |
> | Mean Slopes | 0.34        | 0.18            | 0.21        | 0.55        | 0.19       |
>
> From the results, we can see that GR-DUET adapts faster in residential environments than in non-residential environments. This is likely due to the training environments being predominantly residential (from R2R), introducing a bias that favors residential scenarios. For different instructions, agents adapt fastest to Scene instructions and least effectively to User instructions. Scene instructions often include conversational fillers, which provide more distinct language patterns than the word variations in User instructions, making them easier to adapt to.

---

> ### Author Response · Authors · 2024-11-22
> **Response to Reviewer dHDF - Part 5**
>
> These analyses highlight both the strengths and limitations of GR-DUET’s adaptation mechanism. While it excels in environments with repetitive layouts, it struggles in multi-floor or irregular environments. Future improvements could focus on leveraging dissimilar memories (e.g., between floors) and reducing training biases to enhance adaptation in more diverse scenarios.
>
> We hope these findings and insights guide future research and improvements in the GSA-VLN task.
>
> ---
>
> > ### Question 6. Practical Deployment
>
> While our work is motivated by real-world scenarios, we follow the standard practice in VLN research of addressing scene adaptation problems within simulators as a foundation for future practical deployment. Nonetheless, we believe our proposed method can be effectively deployed in real-world systems, such as robotics or autonomous agents.
>
> To address this concern, we provide a detailed analysis of the computational and memory overhead associated with our GR-DUET. We use the largest environment in GSA-R2R as an example to demonstrate the resource requirements of GR-DUET during inference.
>
> ### **1. Memory Requirements**
>
> - GPU Memory: GR-DUET requires a peak of **4.3 GB** of GPU memory during inference, which is well within the capacity of modern GPUs. For instance, it can be deployed on terminal servers equipped with hardware like the NVIDIA Jetson AGX Orin or similar devices.
>
> - CPU Memory: The method requires at most **5.3 GB** of RAM, which is easily manageable by most modern robotics platforms.
>
> ### **2. Computational Overhead**
>
> - Inference Latency: GR-DUET achieves an average inference latency of **67 milliseconds** per frame, allowing efficient navigation in most real-world environments.
> - Throughput: The system processes **15 frames per second**, which is sufficient for environments where navigation speed is moderate and does not require high-frequency updates, such as indoor environments with static obstacles.
>
> ### **3. Model Characteristics**
>
> - Model Size: The model includes **180 million** parameters, occupying approximately **2.1 GB** of disk space.
> - Computational Complexity: The model has a computational cost of **1.63 GFLOPs** (excluding visual feature extraction), making it feasible for implementation on robots without imposing excessive computational demands.
>
> These metrics demonstrate that GR-DUET is highly practical for deployment in real-time systems. Its resource requirements align with the capabilities of robotics platforms such as TurtleBot2 and LoCoBot, which have been commonly used in previous works \[1-2\]. We have included this discussion in the revised manuscript to highlight the practical applicability of our approach.
>
> &nbsp;
>
> [1] Anderson, Peter, et al. "Sim-to-real transfer for vision-and-language navigation." CoRL, 2021.
>
> [2] Xu, Chengguang, et al. "Vision and Language Navigation in the Real World via Online Visual Language Mapping." CoRL Workshop, 2023

---

> ### Author Response · Authors · 2024-11-25
> **Kind Request for Discussion and Reconsideration of the Score**
>
> We thank Reviewer dHDF for the valuable feedback and insightful suggestions, which have helped us refine and clarify our work. We have carefully addressed all the raised concerns in our response and uploaded a revised manuscript.
>
> We would greatly appreciate it if Reviewer dHDF could provide further feedback. Your input is invaluable to ensuring the quality and clarity of our work. Or, if our responses have satisfactorily resolved the concerns, we respectfully request reconsideration of the score based on the clarifications and improvements provided.

---

> > ### Comment · Reviewer_dHDF · 2024-11-25
> >
> > Thanks to the author for the detailed response. I read the revision carefully and thank the author for the information about memory and language style. However, I still have some concerns.
> >
> > 1. In data of different styles (such as environment and language), the memory method seems to be only applicable to one style, that is, the parameters are accumulated and updated unsupervised in the same environment, but cannot be transferred to other environments, so what is the benefit of training this method out of distribution. Is this a solution to handle different styles? It seems that different environments need to train such a model.
> > 2. Is the adaptation of language style in scene adaptation the main one, because instructions can be easily generated by LLM, or is it a possible solution to directly use LLM to convert these different styles into the same one, and only need to train responses and actions for one language style. And for the style of the environment, will the distribution of visual and spatial objects have a greater impact than language, because the generation and understanding of these in vision and space bring challenges.

---

> > > ### Author Response · Authors · 2024-11-25
> > > **Response to Reviewer dHDF - Part 1**
> > >
> > > Thanks for your valuable feedback. We answer your concerns as follows:
> > >
> > > ---
> > >
> > > > ### Question 1: Different Styles
> > >
> > > Thank you for raising this point. We would like to address your concern from several perspectives:
> > >
> > > ### **1.1 Focus of this Paper**
> > > As highlighted in Line 47 of the manuscript, the focus of our work is on the common scenario where "agents operate in a consistent environment over time." In such cases, both the environment and instruction styles remain relatively stable, often due to fixed instructors or specific use cases. Consequently, the GSA-VLN task, the GSA-R2R dataset, and our GR-DUET method are all designed to address this scenario, aiming to "improve performance over time while executing navigation instructions in a specific scene throughout their lifetime" (Line 74).
> > >
> > > ### **1.2 Transferability of GSA-VLN**
> > > Our work is primarily centered on environment-specific adaptation rather than transferring an adapted model from one environment to another or handling multiple environments simultaneously. These scenarios fall under the domain of transfer learning [1] or continual learning [2], which, while related, are outside the scope of our study.
> > >
> > > Moreover, as noted in Lines 79–81, "Although GSA-VLN focuses on environment-specific adaptation, the agent must also be general enough to adapt to a diverse range of scenes as its target environment, given the wide variety of real-world settings." This means that while the adapted model is **environment-specific**, our adaptation method is **universal** and applicable across different environments and instruction styles, including out-of-distribution (OOD) data. For example, our method can facilitate adaptation whether the agent operates in a home or an office environment, ensuring practicality in varied real-world contexts.
> > >
> > > ### **1.3 Benefits of GSA-VLN**
> > >
> > > **1. Enhanced Performance in the Target Scene**
> > > As shown in Tables 4–6 of the paper, our GR-DUET demonstrates substantial improvements over the vanilla DUET model, achieving an **8%** increase in Success Rate (SR) across all evaluation splits. In specific environments, performance improvements reach as high as **25%**, as illustrated in Figure 10. Such improvements are critical for tasks requiring high reliability and accuracy.
> > >
> > > **2. Unsupervised Adaptation Without Human Involvement**
> > > While our method involves model updates or graph construction tailored to a specific scene, this adaptation is entirely unsupervised and performed autonomously by the agent during its normal navigation. As noted in Lines 75–78, the agent behaves the same as in traditional VLN settings externally, and the adaptation procedure is conducted without additional feedback or assistance. This means the adaptation occurs seamlessly without requiring human intervention or specialized configurations, making the process both practical and realistic for real-world deployment.
> > >
> > > ### **1.4 Conclusion**
> > > The GSA-VLN task is designed to address the persistent environment scenario, ensuring both practicality and significance. It not only enhances performance in target scenes but also does so in an autonomous and unsupervised manner, without introducing additional complexity for the user. While it does not address multi-environment or multi-style scenarios explicitly, it provides a solid foundation for exploring these directions in future research.
> > >
> > > We hope this response clarifies the scope and benefits of our work. Thank you again for your thoughtful feedback!
> > >
> > > &nbsp;
> > >
> > > [1] Qiao, Yanyuan, et al. "VLN-PETL: Parameter-Efficient Transfer Learning for Vision-and-Language Navigation." ICCV. 2023.
> > >
> > > [2] Jeong, Seongjun, et al. "Continual Vision-and-Language Navigation." arXiv preprint arXiv:2403.15049. 2024.

---

> ### Author Response · Authors · 2024-11-25
> **Response to Reviewer dHDF - Part 2**
>
> > ### Question 2: Language vs. Environment
>
> Thank you for raising this important point. Below, we address each aspect of your question in detail.
>
> ### **2.1 Is the adaptation of language style in scene adaptation the main one?**
> We believe both language adaptation and environment adaptation are equally important in scene adaptation, and they are inherently intertwined. While instructions can sometimes be generated or refined independently, their content often reflects the specific characteristics of the environment. In our work, the instructions are not purely "generated" by LLM but rather "refined" based on speaker-generated inputs. This refinement preserves the connection between language and the environment, making absolute isolation of these influences impractical. For example, instructions in a shop often involve specific product categories, such as "head to the produce section.", while instructions in a home commonly reference rooms like the "bedroom" or "kitchen." These contextual differences illustrate how environment and language styles are inherently linked, introducing unique challenges for navigation models. Our results further indicate that the adaptation process must address both linguistic and visual factors. For instance, solely focusing on language or visual adaptation (e.g., using MLM or MRC) does not lead to significantly higher performance gains, highlighting the importance of considering both aspects together.
>
> ### **2.2 Is it a possible solution to directly use LLM to convert these different styles into the same one?**
> Yes, this is a possible solution and we have provided some preliminary experiments to explore its potential in our previous response. We emphasize them again here. We tested an intuitive idea of whether an LLM could translate styled instructions back into a basic style to facilitate the understanding of these styles for navigation models. This was evaluated on the Val-R-Scene split using three sets of instructions:
>
>   1. **Basic:** Instructions after Stage 2.
>   2. **Scene:** Instructions transformed from Basic after Stage 3.
>   3. **Translated:** Scene instructions translated back into Basic style by an LLM.
>
>   The performance of these instruction types is summarized in Table 1.
>
> **Table 1: Performance comparison of instruction styles on the Val-R-Scene split.**
>
> | **Instructions** | **Basic** | **Scene** | **Translated** |
> | ---------------- | --------- | --------- | -------------- |
> | SR               | 46.37     | 42.30     | 44.83          |
>
> The results reveal that LLM-based translation improved performance over Scene instructions but did not fully close the gap with Basic instructions. This limitation arises from the open-vocabulary nature of LLMs, which introduces noise and leads to information loss, thereby reducing the effectiveness of the approach. Since this solution is not environment-specific, it falls outside the scope of scene adaptation and is not included in our work. However, it represents a promising direction for future work to solve the instruction style problem, like fine-tuning a LLM-based translator or adding the translated instructions into the training process of the navigation model.
>
> ### **2.3 Will the distribution of visual and spatial objects have a greater impact than language?**
> Our results suggest that the relative impact of visual/spatial distributions and language styles depends on their variation from the training data:
>  - Environment Adaptation: As shown in Table 4-6 of the paper, our GR-DUET method addresses the environment changes and achieves a performance increase of approximately 8% in Success Rate (SR). This demonstrates the significant impact of visual and spatial variations.
>  - Language Adaptation: Language style changes lead to smaller performance drops (around 3%–5%), as shown in Table 11 of the paper. However, the impact can increase significantly when language styles involve more distinct variations (e.g., highly divergent characters like Moira), as shown in Table 6. In these cases, our GR-DUET achieves smaller performance gains, reflecting the difficulty of adapting to large linguistic distribution shifts.
>
> We think both language and environment styles play critical roles in scene adaptation, and their relative importance depends on specific factors such as the degree of variation from the training data.

---

> > ### Comment · Reviewer_dHDF · 2024-11-25
> >
> > Thanks for your quick reply. Now, if the task is "to focus on adaptability to a particular environment", then it makes sense to me. Both language and environment style play a key role in scene adaptation, but the language style adaptation problem seems to be easier to solve, such as fine-tuning LLM, etc. It's just that the spatial distribution adaptation is more exciting, although it is also inseparable from the language description. It seems to be an interesting problem to explore both together. Thanks again for the rebuttal; it solves most of my confusion. I will change the score to positive and recommend acceptance of this paper.

---

> > > ### Author Response · Authors · 2024-11-25
> > > **Response to Reviewer dHDF**
> > >
> > > Thank you for your thoughtful response and for taking the time to carefully review our rebuttal. We are delighted that our clarifications addressed your concerns and that our focus on adaptability to specific environments makes sense in this context.
> > >
> > > We greatly value your perspective on the challenges of language and environment adaptation. While language adaptation benefits from established techniques like fine-tuning LLMs, we agree that spatial distribution adaptation presents a unique and complex challenge, particularly in its integration with multimodal reasoning. The interaction between environment-specific spatial patterns and the instructions describing them is a fascinating area, and we believe it offers the potential for more innovations in VLN research.
> > >
> > > Your constructive feedback has been instrumental in refining our manuscript, and we deeply appreciate your recognition of our efforts. Thank you once again for your thoughtful input and positive recommendation!

---

### Author Response · Authors · 2024-11-22
**Response to All Reviewers**

We thank all reviewers for their valuable feedback and constructive suggestions on our work. We are encouraged by the recognition of our novelty in proposing the GSA-VLN task for addressing agent adaptation in persistent environments (reviewers t9oJ, dHDF) and the introduction of our diverse GSA-R2R dataset (reviewers J6Xq, J7gj). We are pleased that the effectiveness of our GR-DUET method, especially in out-of-distribution settings, was well received (reviewers dHDF, WxGN). We also appreciate the acknowledgment of our clear and detailed presentation (reviewers WxGN, J6Xq).

We have carefully reviewed all comments and provided detailed responses to each reviewer. A revised manuscript, addressing all feedback and concerns, will be uploaded soon.

We would like to emphasize the main contribution and positioning of this paper. Our work mainly focuses on introducing the novel GSA-VLN task and corresponding dataset, GSA-R2R, which addresses the challenge of VLN agents adapting to a persistent environment that includes both ID and OOD scenarios and diverse instruction types. Thus, the core of our work focuses on identifying the problem, proposing the task setting, generating a dataset with high diversity and quality, and evaluating existing methods under this setting. While we also introduce the novel GR-DUET method, it is primarily positioned as a baseline to provide an initial, feasible solution for this task rather than as a comprehensive solution to the problem. Our goal is to establish a foundation for further exploration and refinement by the research community, encouraging the development of methods that tackle this more realistic and practical setting.

We hope our responses address all concerns effectively and clarify any misunderstandings regarding our work. If there are any further questions or additional feedback, please feel free to reach out, and we will respond promptly. Thank you again for your thoughtful reviews and constructive insights.

---

### Author Response · Authors · 2024-11-24
**Revised Manuscript**

Following the constructive suggestions from the reviewers, we have revised our manuscript to address the feedback and improve the quality and clarity of our work. The revised sections are marked in **red text** throughout the manuscript. Below, we summarize the major changes:

### **1. More Experiments and Results**
We have added new experiments and extended the results to provide deeper insights:
- **Scalability of Memory Mechanism and GR-DUET:** Detailed analysis of computational and memory requirements, demonstrating that GR-DUET scales effectively in larger environments (Section A.5).
- **Practical Deployment Feasibility:** Discussion and evaluation of the feasibility of deploying GR-DUET in real-time systems (Section A.4).
- **Adaptation Efficiency:** Quantitative analysis of GR-DUET's adaptation speed and performance in various environments. (Section A.7)
- **More Baselines:** Inclusion of MapGPT and NavCoT to enrich the comparisons and provide a broader context for LLM-based methods. (Section A.6)

### **2. More Detailed Descriptions and Analysis**
We have expanded and clarified specific sections to address reviewer concerns:
- **Memory Mechanism:** A comprehensive explanation of how the memory bank is implemented, updated dynamically, and utilized for adaptation (Section 3.2).
- **Baseline Model (DUET):** Additional descriptions of DUET's architecture and functionality (Section A.1.1).
- **Human Study:** More details about the participant demographics, evaluation methodology, and examples of the user interface used in the study (Section A.8).
- **Character Selection:** Detailed justification for selecting five characters for User instructions. (Section A.9).

### **3. Related Work Reorganization**
We have reorganized the *Related Work* section to make it more comprehensive:
- **Comparison with Memory-Based and LLM-Based Baselines:** Added discussions on SG-Nav, InstrucNav, MapGPT, NavCoT, and other related works.
- **Comparison with Existing Expanded Scene Works:** Highlighted differences and contributions compared to datasets like HM3D-AutoVLN, ScaleVLN, and YouTube-VLN.
- **Integration of Section 4 Discussion:** Moved discussions of prior works from Section 4 into the Related Work section for better structure and flow.
- **Relation to Lifelong Learning and Test-Time Adaptation:** Added a discussion of how our work relates to and differs from these areas.

We sincerely thank the reviewers for their thoughtful suggestions and constructive feedback. We believe these revisions address the concerns raised and significantly enhance the quality of our manuscript. We hope these changes meet your expectations, and we welcome any additional feedback to further refine our work.

---

### Meta-Review · Area_Chair_96KV · 2024-12-24

**Metareview:**

This paper introduces General Scene Adaptation for Vision-and-Language Navigation (GSA-VLN), a task aimed at training agents to adapt to specific environments while executing navigation instructions and improving their performance over time. To support this task, the authors present GSA-R2R, an expanded dataset based on Room-to-Room (R2R) that includes more diverse environments, instruction styles, and out-of-distribution examples. They also propose Graph-Retained DUET (GR-DUET), a method that uses memory-based navigation graphs and scene-specific training to help agents retain and utilize scene-specific knowledge. The approach achieves strong results across all GSA-R2R benchmarks and highlights some of the key factors that contribute to adaptability in persistent navigation tasks.

The reviewers, quite unanimously, appreciated the clear writing, practical framing of adapting to a persistent environment, and a novel methodology that's been well-tested compared to prior work. The dataset, based on MP3D scenes, was also noted as a helpful contribution. Some reviewers asked about details of the method (GR-DUET), the memory mechanism, dataset diversity, instruction adaptation, related work, and comparisons to baselines and LLM-based methods. Many of these questions were addressed in the rebuttal, and three reviewers updated their scores, leading to unanimous acceptance from the committee.

The AC concurs with the unanimous decision of the committee. Through discussions with reviewers, several clarifications about method details, scalability of the memory mechanism, and useful baselines were added. The AC encourages the authors to ensure these edits are clearly reflected in the final version of the paper.

**Additional Comments On Reviewer Discussion:**

* dHDF raised a few specific concerns. After two rounds of discussions with the authors, the reviewer agreed that most of these were addressed and updated their score from 5 to 6.
* J6Xq started with a lower rating of 3 but increased it to 6 after asking some follow-up questions and reviewing the authors’ clarifications.
* WxGN raised their rating from 5 to 6 after the rebuttal phase, noting improvements in certain aspects of the paper.
* J7gj acknowledged the authors’ responses but kept their original rating of 6.
* The AC finds the authors convincingly addressed the points raised by t9oJ, who had assigned an overall rating of 8.

---

### Decision · Program_Chairs · 2025-01-22

Accept (Poster)